# Targeting fatty acid synthase reduces aortic atherosclerosis and inflammation

Rodrigo Meade[1,10], Dina Ibrahim[1,10], Connor Engel[1], Larisa Belaygorod[1], Batool Arif[1], Fong-Fu Hsu[2], Sangeeta Adak[2], Ryan Catlett[1], Mingzhou Zhou [3], Ma. Xenia G. Ilagan [3], Clay F. Semenkovich [2] & Mohamed A. Zayed [1,4,5,6,7,8,9] ✉

Fatty acid synthase (FAS) is predominantly expressed in the liver and adipose tissue. It plays vital roles in de novo synthesis of saturated fatty acids and regulates insulin sensitivity. We previously demonstrated that serum circulating FAS (cFAS) is a clinical biomarker for advanced atherosclerosis, and that it is conjugated to low-density lipoproteins (LDL). However, it remains unknown whether cFAS can directly impact atheroprogression. To investigate this, we evaluate whether cFAS impacts macrophage foam cell formation – an important cellular process leading to atheroprogression. Macrophages exposed to human serum containing high levels of cFAS show increased foam cell formation as compared to cells exposed to serum containing low levels of cFAS. This difference is not observed using serum containing either high or low LDL. Pharmacological inhibition of cFAS using Platensimycin (PTM) decreases foam cell formation in vitro. In *Apoe*⁻/⁻ mice with normal FAS expression, administration of PTM over 16 weeks along with a high fat diet decreases cFAS activity and aortic atherosclerosis without affecting circulating total cholesterol. This effect is also observed in *Apoe*⁻/⁻ mice with liver-specific knockout of hepatic *Fasn*. Reductions in aortic root plaque are associated with decreased macrophage infiltration. These findings demonstrate that cFAS plays an important role in arterial atheroprogression.

Atherosclerosis is one of the major global underlying causes of cardiovascular disease[1,2]. Its management predominantly revolves around the mitigation of risk factors such as hyperlipidemia with pharmacological therapies that aim to reduce serum circulating lipid levels[3,4]. Despite the efficacy of HMG-CoA reductase inhibitors (statins) and proprotein convertase subtilisin/kexin type 9 (PCSK9) inhibitors in reducing serum low-density lipoprotein (LDL) and total cholesterol levels, their impact on cardiovascular events is confined to a modest range of 30–60%[5–9]. Moreover, even with an effective reduction of total cholesterol using statin monotherapy, individuals contending with cardiovascular co-morbidities continue to be disproportionately susceptible to atheroprogression[10]. This heightened vulnerability translates into a residual major risk of myocardial infarction, stroke, and major peripheral extremity amputations. Recognizing these

continued challenges, the American Heart Association (AHA) and European Society of Cardiology (ESC) have underscored the need to further investigate the underlying causes of atheroprogression and have prioritized the exploration of alternative treatment strategies for this recalcitrant and morbid disease process[11–14].

Lipids such as cholesteryl esters, triglycerides (TGs), and phospholipids, feature saturated fatty acids that wield a significant influence on atheroprogression[1,15]. A mounting body of evidence accentuates the pivotal role of saturated fatty acids within atheromatous plaques, amplifying plaque instability and the heightened risk of atheroma rupture[16]. Similarly, while de novo synthesis of fatty acids is integral to lipid homeostasis, the process is essential for orchestrating the transformation of monocytes into foam cells, which play a fundamental role in perpetuating atheroprogression and

[1]Section of Vascular Surgery, Department of Surgery, Washington University School of Medicine, St. Louis, MO, USA. [2]Metabolism & Lipid Research, Division of Endocrinology, Department of Internal Medicine, Washington University School of Medicine, St. Louis, MO, USA. [3]Department Biochemistry & Molecular Biophysics, Washington University School of Medicine, St. Louis, MO, USA. [4]Veterans Affairs St. Louis Health Care System, St. Louis, MO, USA. [5]Department of Radiology, Washington University School of Medicine, St. Louis, MO, USA. [6]Division of Molecular Cell Biology, Washington University School of Medicine, St. Louis, MO, USA. [7]McKelvey School of Engineering, Department of Biomedical Engineering, Washington University, St. Louis, MO, USA. [8]CardioVascular Research Innovation in Surgery & Engineering Center, Department of Surgery, Washington University School of Medicine, St. Louis, MO, USA. [9]Division of Surgical Sciences, Department of Surgery, Washington University School of Medicine, St. Louis, MO, USA. [10]These authors contributed equally: Rodrigo Meade, Dina Ibrahim. ✉e-mail: zayedm@wustl.edu

exacerbating plaque instability and vulnerability[17]. Consequently, the modulation of fatty acid synthesis within evolving atheroma lesions is thought to influence the progression of atherosclerosis.

Tissue Fatty Acid Synthase (FAS) is an essential soluble 273 kDa intracellular homodimeric enzyme that catalyzes the de novo synthesis of saturated fatty acid through the conversion of acetyl-CoA and malonyl-CoA into palmitate[18–20]. We recently observed that both tissue FAS and serum circulating FAS (cFAS) are elevated in individuals afflicted with severe atherosclerotic cardiovascular disease[21]. Serum cFAS is predominantly produced by the liver and bound to ApoB in LDL particles[22]. Notably, levels of cFAS in serum have a strong correlation with the content of FAS and saturated fatty acids in arterial tissue impacted by atherosclerosis[22]. In macrophages, FAS is also essential for cholesterol trafficking and cellular stress kinase activation[23]. Here we build upon these prior findings to determine whether targeted inhibition of tissue FAS and/or serum cFAS can impact macrophage foam cell formation and alter the course of in vivo atheroprogression.

## Results

### Serum cFAS induces macrophage foam cell formation

Macrophage cytoplasmic lipid droplet accumulation and foam cell formation is a hallmark of atheroprogression[24]. We evaluated whether macrophages conditioned with native human serum either containing high cFAS or LDL can impact foam cell formation in vitro (Fig. 1A, B). Interestingly, we observed a significant correlation between serum cFAS content in the conditioned media and the percentage of macrophages that formed foam cells (Fig. 1C; $R^2 = 0.445$, $p = 0.049$). Conversely, no correlation was observed between serum LDL content and the percentage of macrophage foam cell formation (Fig. 1D; $R^2 = 0.05$, $p = 0.592$).

Macrophages exposed to serum with varying cFAS and LDL levels showed differential foam cell formation (Fig. 1E, F). Notably, high cFAS/low LDL serum (25.84 ng/mL) induced significantly more foam cells than low cFAS/high LDL conditions ($p = 0.008$), evidenced by increased intracellular lipid droplets. Serial dilutions of high cFAS/low LDL serum with FAS-undetectable healthy donor serum were performed to assess the relationship between cFAS levels and macrophage lipid accumulation. Quantitative analysis revealed a significant difference in intracellular lipid droplets between high cFAS/low LDL serum (17.04 ng/mL) and healthy donor serum ($p < 0.0001$). A 50% decrease in relative lipid droplets per cell was observed at a 1:2 dilution ($p = 0.0002$), with a further reduction at a 1:4 dilution ($p = 0.0137$; Supplementary Fig. 1A, B). This dose-dependent response suggests a direct relationship between cFAS levels and macrophage lipid accumulation.

Treatment with the FAS inhibitor PTM (20 µM) significantly decreased macrophage-derived foam cell formation when cells were conditioned with serum containing high cFAS (25.84 ng/mL) and low LDL (<90 mg/dL; Fig. 1G; $p < 0.05$). Furthermore, intracellular FAS activity in macrophages that were conditioned with serum containing high cFAS was higher than in macrophages conditioned with serum containing high LDL (Fig. 1H; $p < 0.05$). Treatment with PTM resulted in a marked reduction of intracellular FAS activity in treated macrophages (Fig. 1H; $p < 0.01$), suggesting that cFAS is a potent factor in foam cell induction. Additionally, FAS-specific inhibitors TVB-2640 (100 nM) and GSK2194069 (100 nM) significantly reduced foam cell formation in differentiated macrophages conditioned with high cFAS/low LDL serum (16.24 ng/mL), compared to untreated cells ($p = 0.003$ and $p < 0.0001$, respectively; Fig. 1I, J). The use of 5% FBS as a negative control confirmed that foam cell formation is specifically induced by FAS activity.

### Intracellular FAS accumulation in macrophages under high cFAS/low LDL

To distinguish between the impact of exogenous and endogenous FAS, we performed FAS staining on macrophages under various conditions. Immunofluorescence analysis of FAS (FAS-Alexa®488) revealed distinct distribution patterns in macrophages exposed to different cFAS/LDL conditions, with and without PTM treatment (Fig. 2A). Quantification of FAS fluorescence intensity showed significant variations across treatment groups (Fig. 2B). Macrophages exposed to high cFAS/low LDL serum exhibited markedly increased FAS fluorescence intensity compared to low cFAS serum conditions in both PTM-treated ($p > 0.05$, $n = 10$) and untreated groups ($p < 0.0001$, $n = 10$). Similar findings were observed using ELISA analysis of FAS content in macrophages that were exposed to high cFAS/low LDL and treated with PTM (Fig. 2C). Differentiated macrophages conditioned with high cFAS/low LDL showed no significant difference in FAS content between PTM-treated and untreated cells. Cell viability assays confirmed that PTM did not significantly affect cell survival at the concentrations used 48 h post-treatment ($p > 0.05$, $n = 3$; Fig. 2D).

### Knockdown and inhibition of FAS alter serum and tissue lipidomics

$Fasn^{+/+}Cre^-Apoe^{-/-}$ and $Fasn^{fl/fl}Cre^+Apoe^{-/-}$ were maintained on a high-fat diet for 16 weeks. A group of $Fasn^{+/+}Cre^-Apoe^{-/-}$ mice also received PTM throughout this period, and serum, liver, and adipose tissue were collected (Fig. 3A). Following 16 weeks of a high-fat diet, $Fasn^{fl/fl}Cre^+Apoe^{-/-}$ mice had significantly less weight gain compared to $Fasn^{+/+}Cre^-Apoe^{-/-}$ mice (51% vs 54% increase in weight; $p < 0.05$; Fig. 3B). On the other hand, $Fasn^{+/+}Cre^-Apoe^{-/-}$ mice treated with PTM had no significant change in body weight compared to untreated $Fasn^{+/+}Cre^-Apoe^{-/-}$ mice (Fig. 3B).

Compared to $Fasn^{+/+}Cre^-Apoe^{-/-}$ mice, $Fasn^{fl/fl}Cre^+Apoe^{-/-}$ mice treated with and without PTM, demonstrated no significant differences in liver and adipose TGs at 16 weeks (Fig. 3C, D). However, free fatty acids (FFAs) were notably decreased in the liver (Fig. 3E), and significantly elevated in adipose tissue of $Fasn^{+/+}Cre^-Apoe^{-/-}$ mice that received PTM treatment (Fig. 3F; $p < 0.01$).

After initiation of a high-fat diet, $Fasn^{fl/fl}Cre^+Apoe^{-/-}$ mice demonstrated a different pattern of FAS content and activity in the hepatic and adipose tissue. As expected, $Fasn^{fl/fl}Cre^+Apoe^{-/-}$ mice demonstrated a significant decrease in FAS content and activity in hepatic tissue (Fig. 3G, H; $p < 0.05$). Interestingly, $Fasn^{fl/fl}Cre^+Apoe^{-/-}$ mice demonstrated a significant increase in FAS content in white adipose (Fig. 3I; $p < 0.01$). PTM treatment did not impact FAS content in the liver but led to a significant decrease in FAS activity in white adipose (Fig. 3J; $p < 0.05$). Similarly, there was a numerical, but not statistically significant, reduction in FAS activity in hepatic tissue of PTM-treated $Fasn^{+/+}Cre^-Apoe^{-/-}$ mice (Fig. 3H; $p = 0.69$).

### Conditional liver Fasn knockdown and FAS inhibition impact serum cFAS and tissue FAS content and activity

We observed a significant decrease in serum cFAS content in $Fasn^{fl/fl}Cre^+Apoe^{-/-}$ mice before the initiation of a high-fat diet regimen (Fig. 4A; $p < 0.01$). Similarly, PTM treatment significantly reduced cFAS content in $Fasn^{fl/fl}Cre^-Apoe^{-/-}$ mice (Fig. 4A; $p < 0.05$). Interestingly, serum cFAS activity was reduced in $Fasn^{fl/fl}Cre^+Apoe^{-/-}$ mice, and PTM-treated $Fasn^{+/+}Cre^-Apoe^{-/-}$ mice (Fig. 4B). This reduction was more significant after 9 and 16 weeks with a high-fat diet (Fig. 4C, D; $p < 0.01$). All mouse groups demonstrated hypercholesteremia with total cholesterol >1400 mg/dL at 16 weeks (Fig. 4E). We also observed a significant increase in serum TG levels across all groups, particularly by week 3 of the intervention, followed by a slight decline at weeks 6 and 14 (Fig. 4F). Similarly, serum FFA showed a significant rise by week 3, with a subsequent decrease observed by week 6, and continuing to decline through week 14, but not statistically different between mouse groups (Fig. 4G). Interestingly, serum glucose levels peaked at week 6 and showed a modest reduction by week 14 (Fig. 4H).

### Fasn conditional knockdown or pharmacological inhibition reduces atheroprogression

We next evaluated arterial atheroprogression in $Fasn^{fl/fl}Cre^+Apoe^{-/-}$ mice and PTM-treated $Fasn^{+/+}Cre^-Apoe^{-/-}$ mice that were maintained on a high-fat diet for 16 weeks and aortas harvested for analysis (Fig. 5A). Compared to $Fasn^{+/+}Cre^-Apoe^{-/-}$, $Fasn^{fl/fl}Cre^+Apoe^{-/-}$ mice demonstrated a significant reduction in total aortic atherosclerotic plaque formation

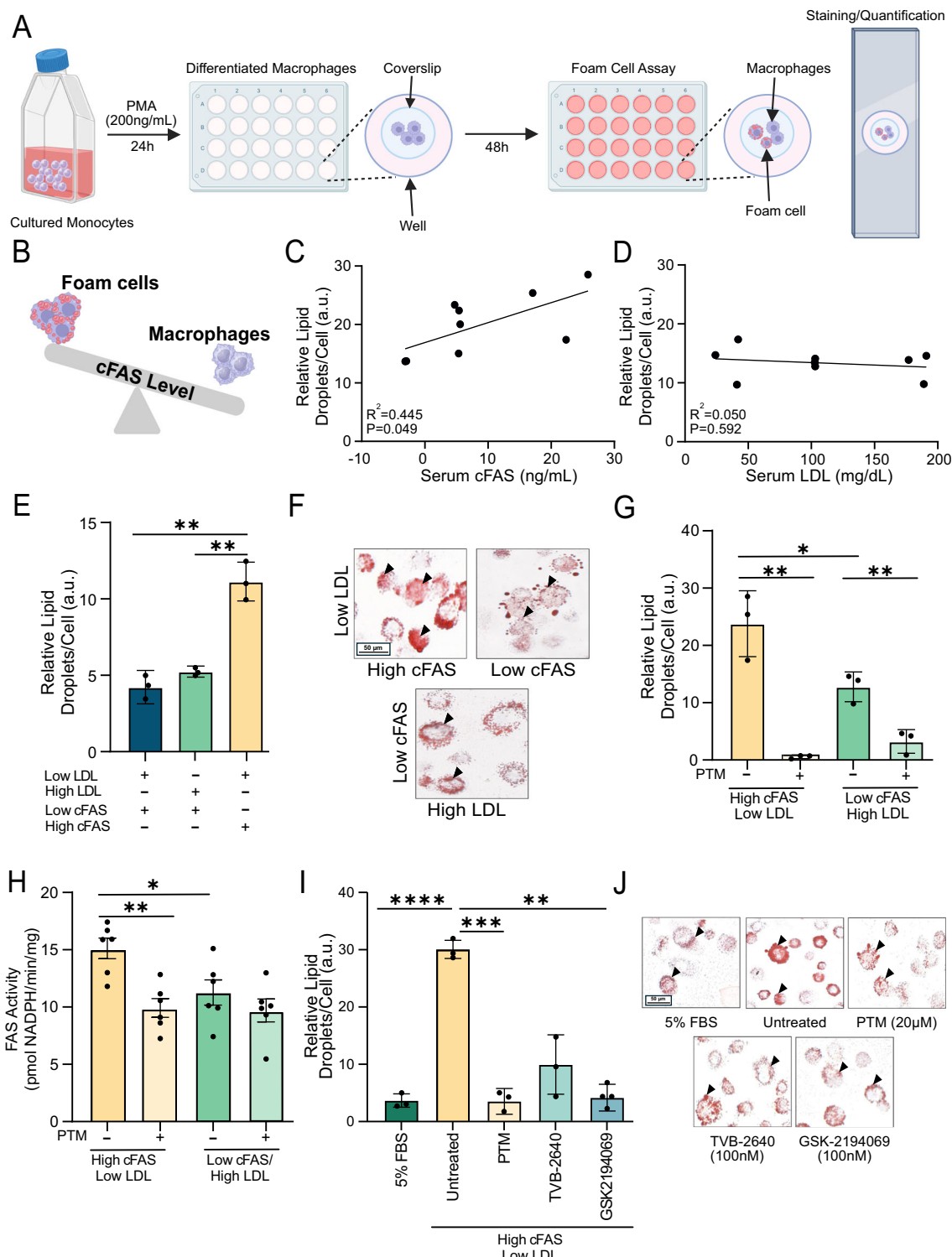

**Fig. 1 | Differential impact of cFAS and LDL on macrophage foam cell formation in vitro. A** Schematic representation of in vitro experiments used to evaluate the impact of cFAS or LDL on macrophage foam cell formation. **B** Correlation between the percentage of Oil Red O-positive foam cells and serum concentrations of cFAS and LDL. **C** Evaluation of foam cell formation in differentiated macrophages exposed to varying concentrations of cFAS and **D** LDL levels; n = 9. The percentage of foam cells was quantified for each condition. **E** Assessment of foam cell formation in differentiated macrophages under different conditions: high cFAS/low LDL (25.84 ng/mL) and low cFAS/high LDL, with or without treatment with PTM (20 μM). **F** Representative Oil Red O staining of lipid droplets in macrophages under different cFAS and LDL conditions. Increased lipid droplets are visible under high cFAS and high LDL conditions, compared to low cFAS conditions. **G** Effect of PTM

treatment (20 μM) on lipid droplets in differentiated macrophages under high cFAS/low LDL and low cFAS/high LDL conditions, and **H** FAS activity. **I** Differentiated macrophages conditioned with high cFAS/low LDL serum (16.24 ng/mL) were treated with different FAS inhibitors: PTM (20 nM), TVB-2640 (100 nM), and GSK2194069 (100 nM). Untreated cells served as a positive control, and cells cultured in 5% FBS served as a negative control. **J** Oil Red O staining showed the reduction of lipid droplets in macrophages treated with PTM, TVB-2640, and GSK2194069 compared to controls (5% FBS). Scale bar: 50 μm. Arrows indicate lipid droplets within macrophages. Data are presented as mean ± SEM (n = 3 independent experiments). *p < 0.05, **p < 0.01 compared to control (untreated) conditions. The illustrations in (**A**) (https://BioRender.com/d25g720) and (**B**) (https://BioRender.com/e26i107) were created using BioRender.

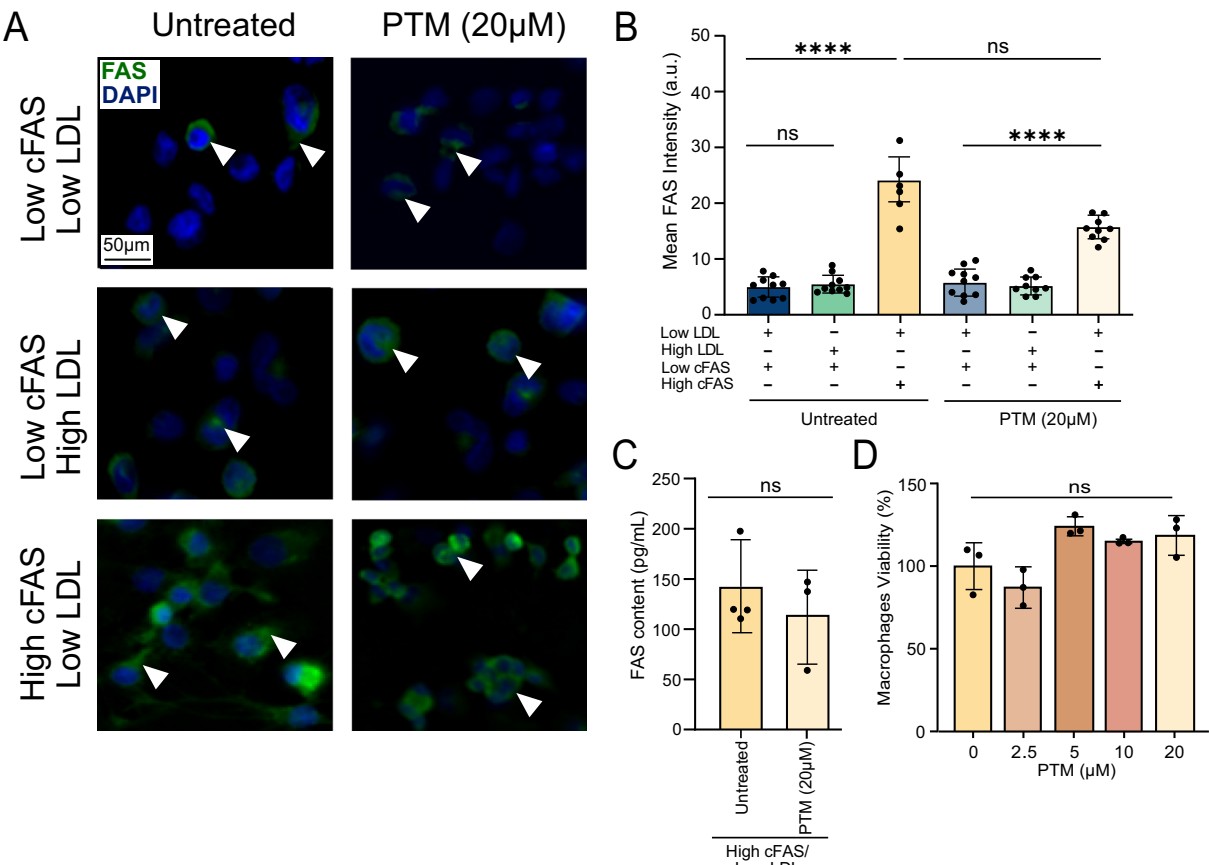

**Fig. 2 | Impact of conditioned serum on intracellular and extracellular FAS in macrophages.** **A** Immunofluorescence of FAS (FAS-Alexa®488) in macrophages under various cFAS/LDL conditions ± PTM treatment. **B** Quantification of FAS fluorescence intensity ($n = 10$). **C** ELISA of FAS content in macrophages treated with high cFAS/low LDL ± PTM, compared to 5% FBS control ($n = 3$). **D** Percentage of cell viability after 48 h of exposure to various PTM concentrations ($n = 3$). Scale bar, 50 µm. Data are mean ± SEM ($n = 3$ independent experiments). $*p < 0.05$, $**p < 0.01$, $***p < 0.001$; ns not significant.

(Fig. 5A, F; $p < 0.001$). This difference was evident in all aortic segments, including the innominate artery plaque ($p < 0.0001$), aortic arch ($p < 0.01$), thoracic aorta ($p < 0.01$), and infrarenal aorta ($p < 0.001$) (Fig. 5B–F). Similarly, PTM-treated $Fasn^{+/+}Cre^-Apoe^{-/-}$ mice also demonstrated significantly reduced total aortic (Fig. 5A, F; $p < 0.01$), as well as reduced plaque in the innominate artery plaque ($p < 0.01$) aortic arch ($p < 0.01$), thoracic ($p < 0.05$) and infrarenal aortic segments ($p < 0.01$) (Fig. 5B–F).

Atherosclerotic plaque formation at the aortic valve roots was also evaluated after 16 weeks of a high-fat diet regimen and with and without PTM treatment. $Fasn^{fl/fl}Cre^+Apoe^{-/-}$ mice and PTM-treated $Fasn^{+/+}Cre^-Apoe^{-/-}$ mice demonstrated significantly reduced aortic valve root plaque formation (Fig. 6A, B; $p < 0.01$). Similarly, $Fasn^{fl/fl}Cre^+Apoe^{-/-}$ mice and PTM-treated $Fasn^{+/+}Cre^-Apoe^{-/-}$ mice demonstrated reduced CD68+ macrophages in the aortic valve wall, while $Fasn^{+/+}Cre^-Apoe^{-/-}$ mice had higher CD68 content in the valve wall atheroma (Fig. 6C; $p < 0.05$).

**Conditional knockdown of *Fasn* or inhibition of FAS alters tissue inflammation**

As expected, hepatic tissue of $Fasn^{fl/fl}Cre^+Apoe^{-/-}$ mice had diminished FAS immunostaining (Fig. 7A, B; $p < 0.05$), and reduced CD68+ macrophage content (Fig. 7A, C; $p < 0.05$). PTM-treated $Fasn^{+/+}Cre^-Apoe^{-/-}$ mice also demonstrated a tendency towards a reduction in liver FAS (Fig. 7A, B; $p = 0.088$), and significantly reduced CD68 content (Fig. 7A, C; $p < 0.0001$).

Interestingly, in white adipose tissue, liver $Fasn^{fl/fl}Cre^+Apoe^{-/-}$ mice demonstrated increased adipocyte area (Fig. 7D, E; $p < 0.05$) and increased FAS content (Fig. 7D, F; $p < 0.001$), but no change in CD68 content (Fig. 7D, G). In PTM-treated $Fasn^{+/+}Cre^-Apoe^{-/-}$ mice, there was a

significant decrease in adipocyte area (Fig. 7D, E; $p < 0.05$), a strongly significant decrease in FAS content (Fig. 7D, F; $p < 0.05$), and a significant decrease in CD68 content (Fig. 7D, G; $p < 0.05$).

## Discussion

Our study evaluates the role of tissue FAS and serum cFAS on atheroprogression and tissue inflammation. We observed a significant increase in macrophage foam cell formation when conditioned with serum containing higher cFAS content. Treatment with FAS-specific inhibitors PTM, GSK2194069, and TVB-2640 significantly blunted foam cell formation. Similarly, in vivo, conditional knockdown of FAS in the liver, or treatment with PTM, greatly reduced aortic atherosclerotic plaque volume and macrophage content in aortic plaque regions. We also observed that FAS targeting impacted liver-adipose tissue crosstalk. Remarkably, although $Fasn^{fl/fl}Cre^+Apoe^{-/-}$ mice exhibited hypercholesteremia while maintaining a 42% high-fat diet, they developed minimal aortic atherosclerotic plaque. Overall, these findings highlight the indispensable roles that tissue FAS and serum cFAS contribute to atheroprogression.

Dyslipidemia is a known risk factor for atheroprogression and cardiovascular disease[15,25–27]. Individuals with familial hypercholesterolemia are born with dramatically elevated serum LDL cholesterol, develop early atherosclerotic disease onset, and are at higher risk of cardiovascular complications if not intensively treated[28]. Lipid-lowering medications, such as statins (coenzyme A reductase inhibitors), fibrates, and PCSK9 inhibitors, are first-line in the management of hyperlipidemia and aim to reduce serum circulating lipids thereby reducing the risk of cardiovascular events such as myocardial infarction (MI), stroke, and significant lower extremity amputations resulting from peripheral arterial occlusive disease[29–31]. However,

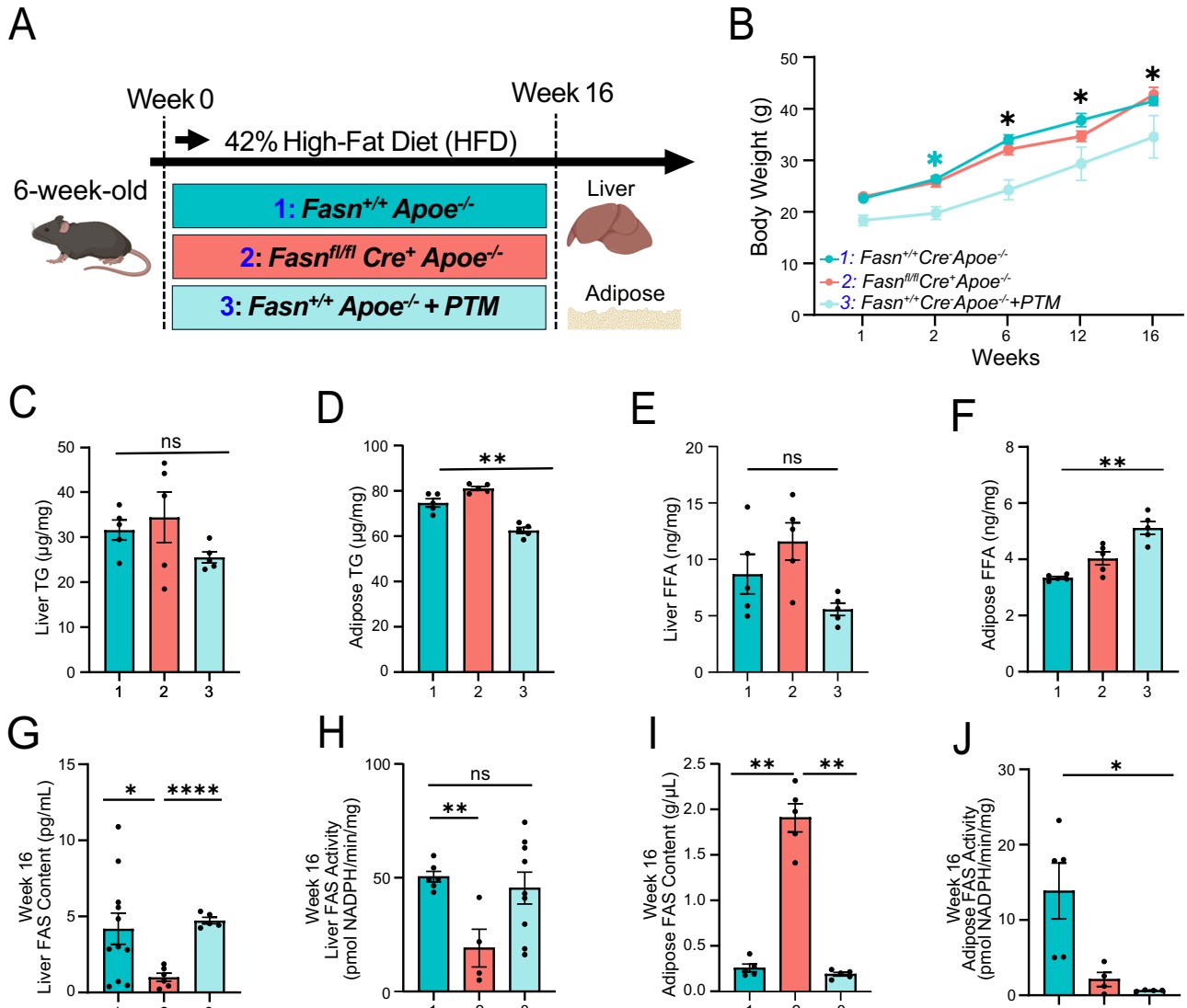

**Fig. 3 | FAS inhibition affects murine lipid homeostasis. A** Schematic representation of in vivo murine experiments using *Fasn^+/+Cre^−Apoe^−/−* (1) and *Fasn^fl/fl Cre^+Apoe^−/−* (2) that were maintained on a high-fat diet for 16 weeks. A group of *Fasn^+/+Cre^+ Apoe^−/−* mice also received PTM (3) throughout this period. Liver and white adipose tissue were collected from all mouse groups. **B** Impact of *Fasn* knockdown of on murine body weight (*n* = 5 per group) over a 16-week period.

**C** and **D** Liver and white adipose tissue triglycerides for each murine group (*n* = 6 per group). **E** and **F** Liver and white adipose tissue non-esterified free fatty acid content (*n* = 6 per group). **G** and **H** FAS content and activity in liver tissue (*n* = 6 per group). **I** and **J** FAS content and activity in adipose tissue (*n* = 6 per group). *\*p* < 0.05, *\*\*p* < 0.01. Data are mean ± SEM. Illustration in (**A**) was created using BioRender (https://BioRender.com/u19d713).

despite the reduction of LDL with statins (20–50%) and PCSK9 inhibitors (50–65%), cardiovascular events are still only reduced by 30–60% in patients who are treated with medications within these drug categories[32,33]. This leaves most individuals with a significant residual risk of major cardiovascular events and an unclear management strategy to reduce cardiovascular morbidity and mortality[5,34]. These persistent clinical gaps have contributed to a growing suspicion that there are likely additional key contributors to atheroprogression that are yet to be identified and therapeutically targeted.

While individuals with high serum LDL levels (>190 mg/dL) are known to have a higher incidence of MI and stroke, this is indeed not always the case[35]. For example, in The Multi-Ethnic Study of Atherosclerosis (MESA), which evaluated >23,000 patients over a 16-year period, high serum LDL was observed to not be a risk factor for the incidence of atherosclerotic cardiovascular disease in individuals who had a zero coronary artery calcium (CAC) score on CT angiography[36]. Similarly, in a study of >136,000 patients who were hospitalized for an acute MI, it was observed that nearly 75% of patients had serum LDL levels that would indicate they were not at high risk of cardiovascular events[37]. These studies highlight that

beyond LDL cholesterol there are additional serum and/or tissue lipid mediators that can influence whether a patient is either at higher or lower risk for atherosclerotic disease progression.

Fatty acids are essential lipids that serve as functional components for TGs, phospholipids, and cholesterol esters. These lipid mediators impact a diverse array of cellular and tissue processes, including cell membrane structure and integrity, as well as serving as biological energy storage units during catabolism[38]. On the other hand, dysregulation of fatty acid synthesis contributes to deleterious conditions such as obesity, non-alcoholic fatty liver disease, and type 2 diabetes[39–41]. In macrophages, fatty acids play key roles in cholesterol uptake, esterification, and lipid efflux[17,42]. However, dysregulation of fatty acid synthesis is known to impact macrophage function, polarization, and phenotypic transformation[16,20,23]. Fatty acid synthesis impacts macrophage cholesterol efflux and foam cell formation[42,43]. This is of particular importance since foam cell accumulation in the arterial intima has been linked to arterial wall atheroma progression and plaque vulnerability[16,24,44,45]. Similarly, fatty acid metabolism participates in the transition of vascular smooth muscle cells to macrophage-like cells in

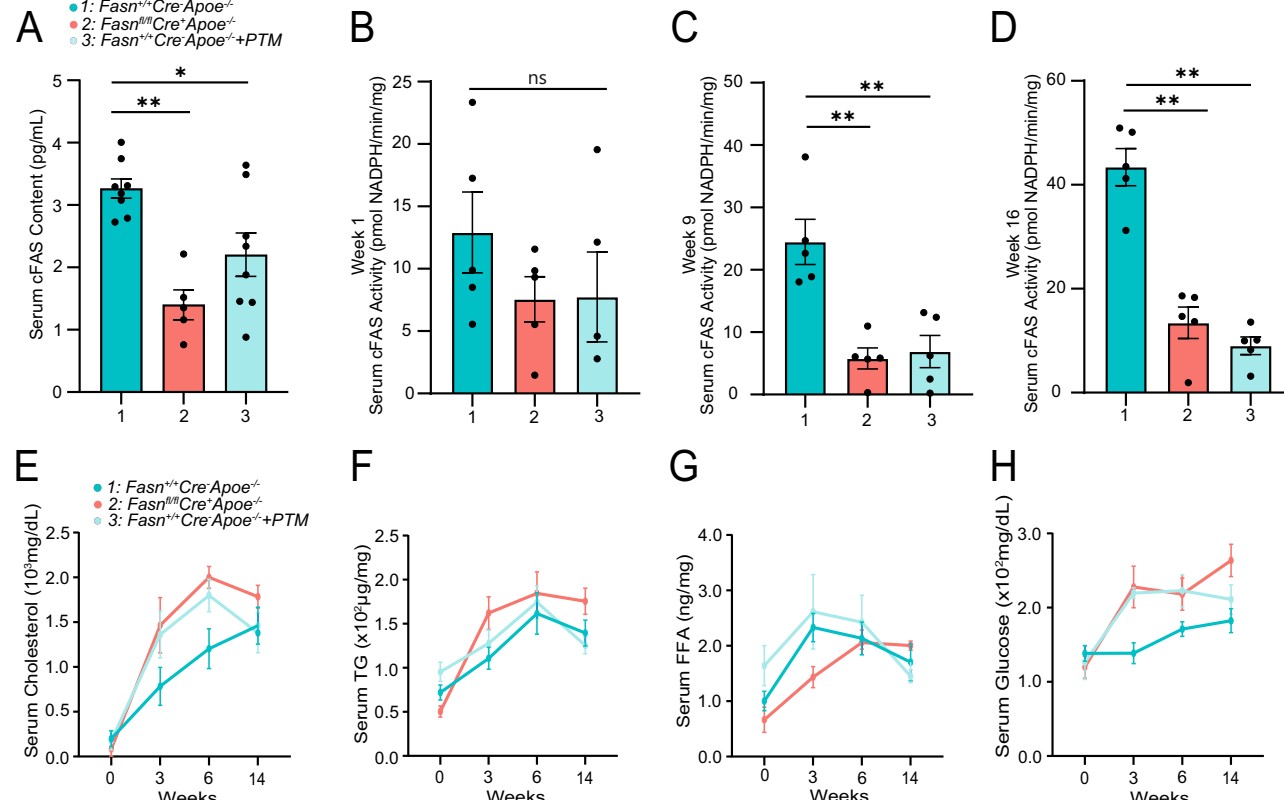

**Fig. 4 | Conditional liver-specific knockdown of *FAS* and PTM treatment impacts serum cFAS content and activity.** Serum specimens from *Fasn^+/+ Cre^- Apoe^-/-* mice, *Fasn^fl/fl Cre^+ Apoe^-/-* mice, and *Fasn^+/+ Cre^- Apoe^-/-* mice treated with PTM, were analyzed for cFAS content (**A**) and cFAS activity (**B**–**D**; n = 5 per mouse group). **E** and **F** The supernatant of the digested liver was evaluated for cFAS content and activity (n = 5). **G** and **H** The supernatant of digested white adipose tissue was evaluated for cFAS content and activity (n = 5). Data are mean ± SEM. *p < 0.05, **p < 0.01, ***p < 0.001.

atherosclerotic lesions[16,46,47]. Here we demonstrate that serum cFAS plays an important role in macrophage foam cell formation and that both serum cFAS and endogenous liver FAS play an essential role in aortic atherosclerosis.

We previously demonstrated that conditional knockdown of *Fasn* in the liver, but not in skeletal muscle, leads to reduced serum cFAS. Additionally, we observed that cFAS co-immunoprecipitated with ApoB in LDL cholesterol serum fractions[22]. These findings previously led us to conclude that cFAS is produced by the liver and is released into the bloodstream bound to ApoB in lipoproteins such as LDL. Given the relative concentrations of LDL and cFAS in human serum it is evident that cFAS concentrations are at least an order of magnitude less than those for LDL. Thus, while cFAS may serve as cargo attached to the highly diverse proteome of LDL, not all LDL particles carry cFAS. This is presumably why we observed that human serum had variable content of cFAS and LDL. In our biobanked samples, some samples had higher cFAS content (>16 ng/mL), and others essentially had undetectable cFAS. Naturally, we also observed serum samples that had very high LDL (>180 mg/dL), while others that had low LDL (<90 mg/dL). Since we previously reported that there was no correlation between cFAS and LDL content in human serum, we intentionally evaluated the impact of human serum samples with either high and low cFAS or LDL[21]. Like others who demonstrated that LDL alone does not cause macrophage foam cell formation, we also observed that macrophages conditioned with serum containing high LDL, but low cFAS, did not lead to foam cell formation[45,48–50].

The mechanistic process that facilitates cFAS impact on foam cell formation is currently not fully elucidated. However, prior work demonstrates that endogenous FAS in macrophages is essential for retaining plasma membrane cholesterol, cellular adhesion, and migration, as well as

recruitment into adipose tissue that facilitates chronic tissue inflammation induced by nutrient-dense diets[42,51–55]. Macrophage-specific FAS deficiency exhibited a reduced inflammatory response, characterized by decreased expression of pro-inflammatory cytokines (TNF-α, IL-1β). Notably, these FAS-deficient macrophages exhibited upregulation of *Abca1*, which promotes the removal of excess cholesterol, potentially contributing to the reduction of cell formation and atheroprogression[42]. In our study, we similarly observed that pharmacological inhibition of FAS with PTM in mice maintained on a 42% high-fat diet dramatically reduced macrophage infiltration in both hepatic and white adipose tissue (Fig. 7C, F). While conditional knockdown of *Fasn* in liver tissue also reduced hepatic macrophage infiltration, it did not have as robust of a phenotype in white adipose tissue. Moreover, compared to the conditional knockdown of *Fasn*, treatment with PTM had a more dramatic reduction of macrophages in the liver (127% difference) and white adipose (96% difference) tissue, suggesting that its inhibition of serum cFAS was likely playing a major role in these findings. Liver-specific *Fasn* knockout leads to the accumulation of de novo lipid species that activate PPARα, a crucial regulator of metabolism[56] PPARα activation helps maintain glucose, lipid, and cholesterol homeostasis in *Fasn*-deficient livers. These findings highlight that FAS inhibition may trigger complex compensatory mechanisms that can impact metabolism in several ways.

While global *Fasn* deficiency is embryologically lethal, acute pharmacological inhibition of FAS is a topic of multiple prior investigations, particularly since FAS is elevated in malignant tissue, and serum cFAS is also elevated in individuals with certain metastatic tumors[57–60]. Indeed, there are currently FAS inhibitors that are undergoing efficacy testing in phase II human clinical trials and are demonstrating promise[61]. Due to PTM availability and potent inhibition in our in vitro studies, PTM was utilized for the in vivo experiments. PTM is a commonly used FAS

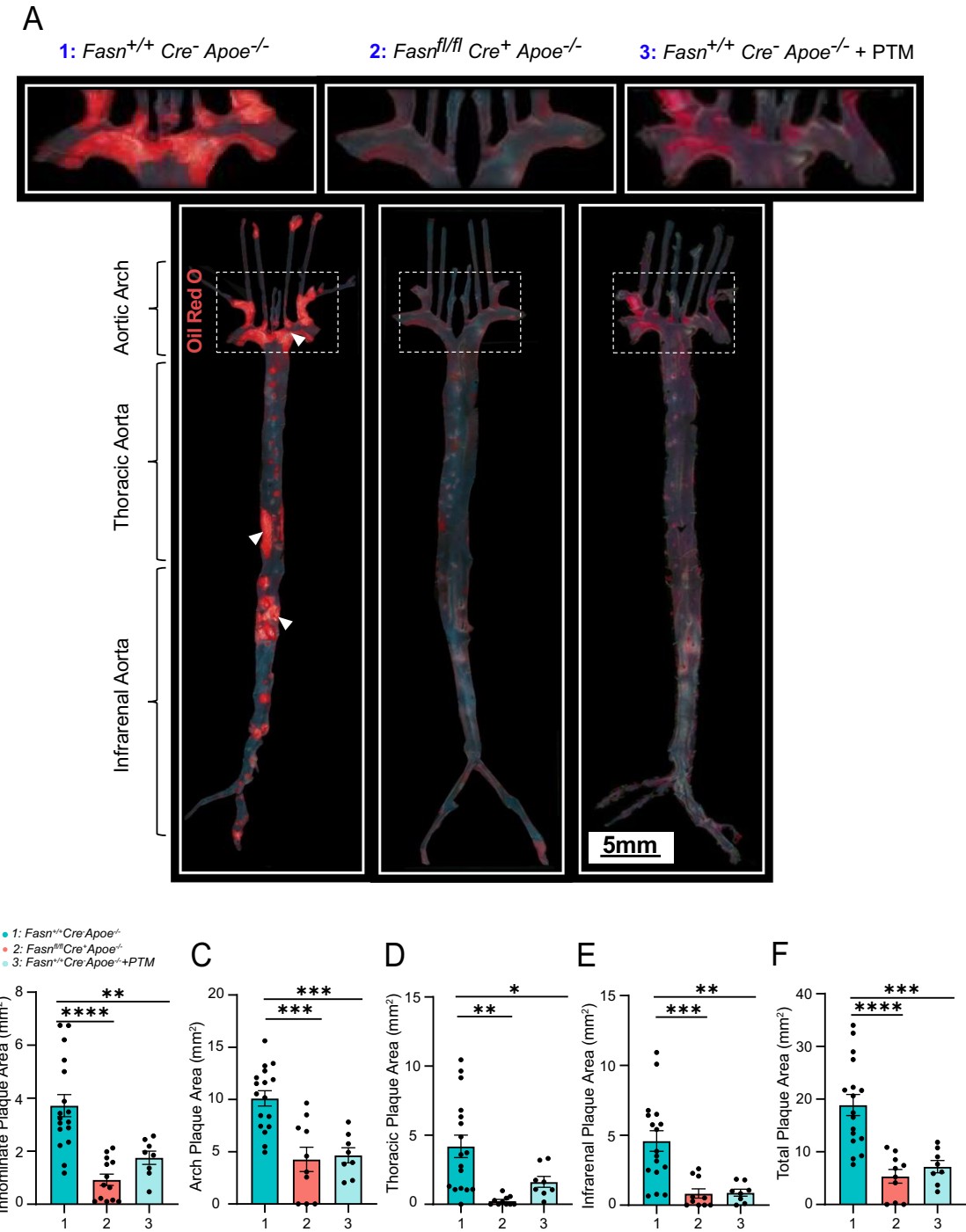

**Fig. 5 | Targeting FAS reduces the aortic atherosclerotic plaque burden.**
**A** Representative enfacements of aortic specimens from different mouse groups that were stained with Oil Red O. Plaque areas are visualized in red. **B** Total aortic plaque assessment in *Fasn*$^{+/+}$ *Cre*$^-$ *Apoe*$^{-/-}$ ($n = 17$), *Fasn*$^{fl/fl}$ *Cre*$^+$*Apoe*$^{-/-}$ ($n = 10$), and *Fasn*$^{+/}$ $^+$*Cre*$^-$*Apoe*$^{-/-}$ that were treated with PTM ($n = 8$). **C** Plaque burden in the aortic arch segment, **D** thoracic aorta, **E** infrarenal aorta, and **F** innominate artery. Data are mean ± SEM. *$p < 0.05$, **$p < 0.01$, ***$p < 0.001$.

inhibitor that is naturally derived from *Streptomyces platensis* bacteria. It selectively and competitively binds to both bacterial and mammalian FAS and forms stable complexes with FAS subunits[62]. While its impact on microbiota is unknown, in *db/db* mice, PTM inhibits de novo fatty acid synthesis and enhances glucose oxidation[63]. Consistent with the findings, we observed that PTM treatment of *Fasn*$^{+/+}$ *Cre*$^-$ *Apoe*$^{-/-}$ mice supported normal weight gain suggesting acute non-lethal dosing. Prior studies suggest that this phenotype is observed due to improved hepatic glucose uptake and glycolysis[63]. However, our study clearly demonstrates

that PTM also impacts white adipose FAS content and activity, as well as adipocyte lipid storage (expressed as adipocyte area). This may reflect a compensatory mechanism and tissue-specific metabolic adaptation where adipose tissue increases FFAs uptake and storage in response to systemic FAS inhibition, potentially through upregulation of fatty acid transporters and altered lipolysis regulation. Importantly, this observation suggests that adipose tissue may serve as a lipid buffer, compensating for decreased FAS content and activity in the liver by increasing its capacity to store lipids[64].

The remarkable crosstalk between liver and adipose tissue was not only limited to mice treated with PTM but was also observed in *Fasn^{fl/fl} Cre^- Apoe^{-/-}* mice, which after 16 weeks of a high-fat diet regimen demonstrated significantly elevated FAS content and sustained serum FFA levels. Interestingly, FAS inhibition was previously demonstrated to improve non-alcoholic fatty liver diseases and non-alcoholic

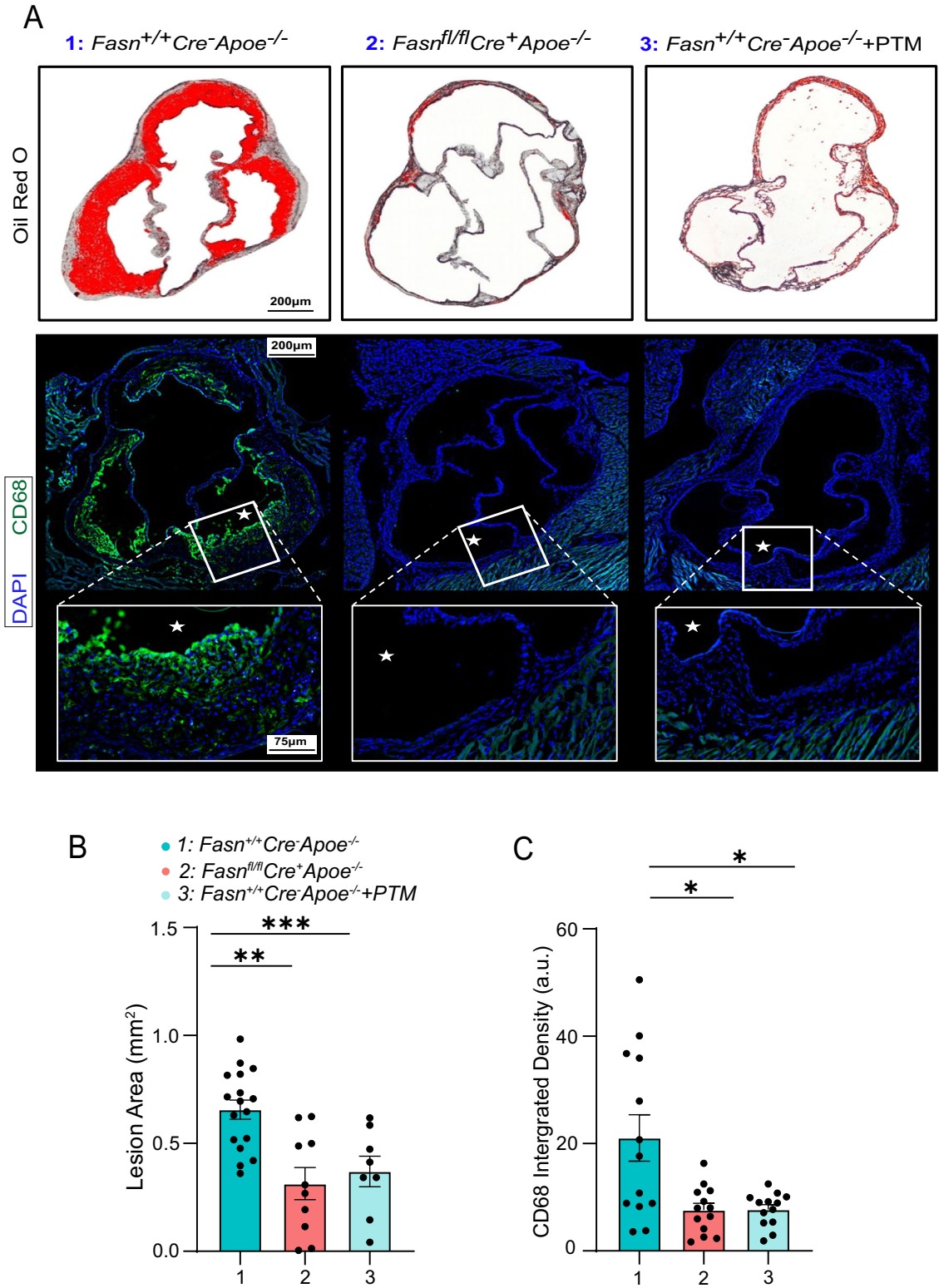

**Fig. 6 | Targeting FAS reduced the aortic root plaque burden. A** Hearts were isolated from *Fasn^{+/+}Cre^- Apoe^{-/-}* mice, *Fasn^{fl/fl} Cre^+ Apoe^{-/-}* mice, and *Fasn^{+/+}Cre^- Apoe^{-/-}* mice treated with PTM, that were maintained on a high-fat diet for 16 weeks. Aortic valve leaflets were sectioned at 10 μm and stained with Oil Red O. The Plaque area is visualized in red. Aortic valve leaflet sections were stained with the macrophage marker CD68 (green) and DAPI nuclear stain (blue). * Indicates lumen. **B** Plaque lesion area percentage was evaluated in each mouse group (*n* = 6). **C** Integrated density was analyzed to evaluate CD68 content in the aortic valve sections of each mouse group (*n* = 6). Data are mean ± SEM.

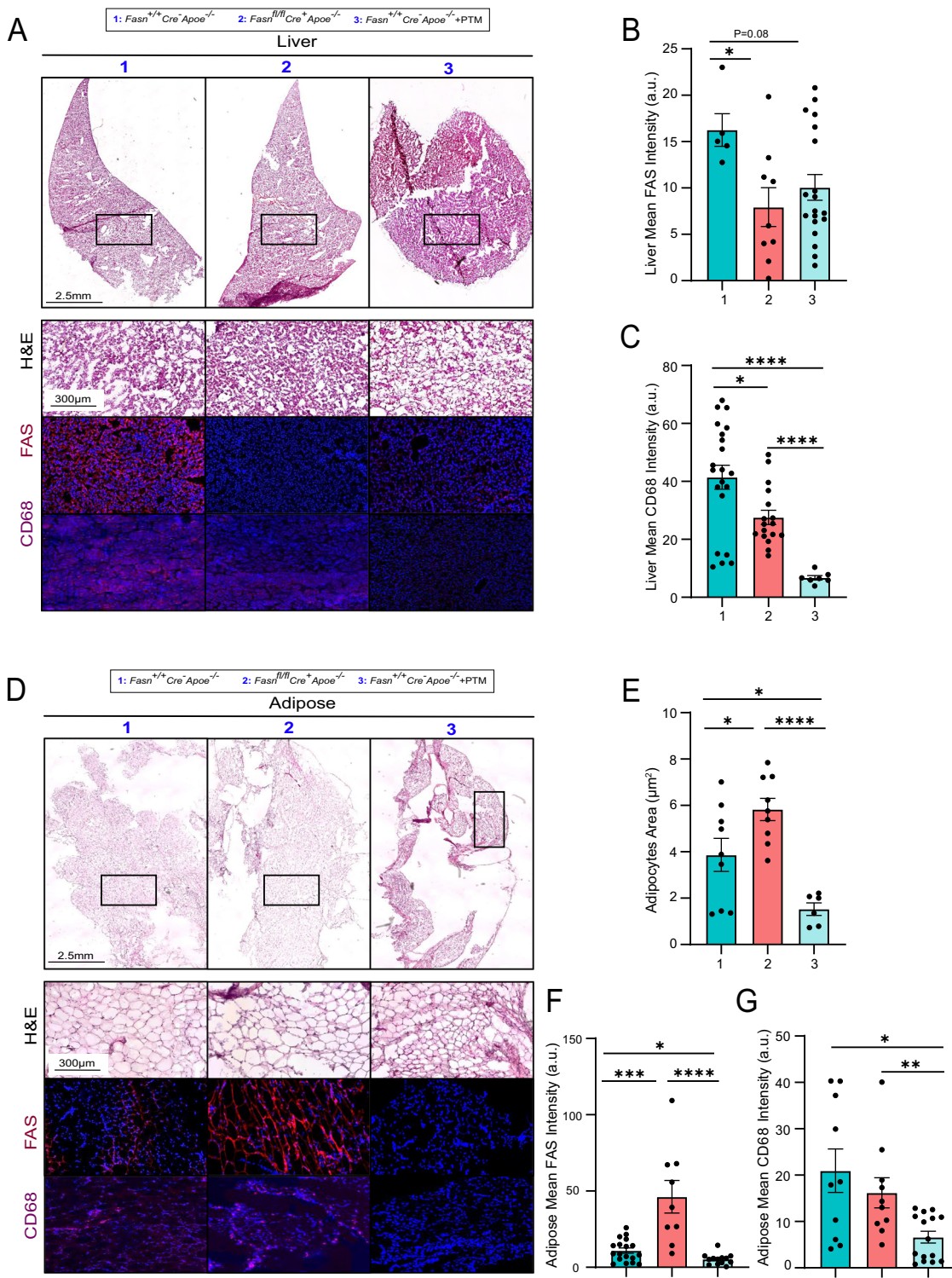

**Fig. 7 | FAS inhibition reduces tissue FAS and inflammation response. A** Liver tissue were collected from *Fasn⁺/⁺ Cre⁻ Apoe⁻/⁻* mice, *Fasn^fl/fl Cre⁺ Apoe⁻/⁻* mice, and *Fasn⁺/⁺ Cre⁻ Apoe⁻/⁻* mice treated with PTM, that were maintained on a high-fat diet for 16 weeks. Tissues were then sectioned and stained with H&E, and immunostained for FAS and CD68. **B** Quantification of liver FAS staining (*n* = 5), and **C** liver CD68 staining (*n* = 5). **D** White adipose tissue were also collected from mouse groups and stained with H&E, and immunostained for FAS and CD68. **E** Average adipocyte vacuole area (*n* = 5), **F** FAS staining (*n* = 5), and **G** CD68 staining (*n* = 5). Data are mean ± SEM.

steatohepatitis by reducing inflammatory markers IL-1β and reducing transaminitis (elevated serum ALT and AST)[60,65]. These findings underscore the importance of considering tissue-specific metabolic adaptations when developing therapies targeting lipid metabolism. Liver and adipose signaling in relation to fatty acid synthesis and macronutrient metabolism has been reported extensively and is a highly orchestrated process[64]. In humans, dietary nutrients, and de novo lipid synthesis in these organ tissue are thought to influence obesity and fatty liver disease[66]. Our findings suggest that cFAS may in part be a vehicle of communication between the liver and white adipose tissue.

## Methods

### Human serum

Native human serum was obtained from the Washington University in St. Louis institutional review board (IRB)-approved vascular biobank[67]. As previously described[68], fresh human serum aliquots were collected from study participants who fasted at least 12 h and concentrated using 100 kDa ultrafiltration centrifuge tubes (Thermo Fisher Scientific, Waltham, MA), at 15,000 g, for 15 min. A resultant minimum volume of 250 μL of concentrated serum was collected for each patient. Concentrated serum samples were stored in aliquots at -80°C for subsequent use in vitro macrophage foam cell experiments.

Serum used for conditioned media studies were obtained from male individuals with similar age demographics and contained either LDL content at low (<90 mg/dL), medium (90–180 mg/dL), or high (>180 mg/dL)[68], and undetectable levels of cFAS (Table 1). Alternatively, media was also conditioned with serum containing either low undetectable (0 ng/mL), medium (5–6 ng/mL), or high (>16 ng/mL) cFAS content, and low levels of LDL (<90 mg/dL; Table 1). Serum LDL content was determined by the Washington University in St. Louis Core Laboratory for Clinical Studies (CLCS), utilizing an N-Geneous® LDL cholesterol kit (Sekisui Diagnostics, #7120) using a Roche Cobas c501 analyzer. Serum cFAS content was determined using commercial ELISA according to the manufacturer's instructions (Aviva, OKEH04869)[21,22,69].

### Tissue culture and foam cell formation assessment

Human U-937 cells (ATCC #CRL-1593.2) were cultured for 48 h in 10% FBS RPMI and then differentiated to macrophages in 5% FBS RPMI treated with 200 ng/ml Phorbol 12-myristate 13-acetate (PMA), 95% (Thermo Fisher, #J63916.MB). Monocytes were allowed 24 h to differentiate into macrophages. Differentiated macrophages were then washed 3× with PBS and incubated for 48 h in cell culture media RPMI (Thermo Fisher, #11875093) conditioned with 5% of human serum containing either low, medium, and high levels of cFAS or LDL, or healthy human donor serum on fibronectin pre-treated coverslips (Table 1 & Fig. 1A). Similarly, macrophages conditioned with serum containing either higher levels cFAS or LDL, were also simultaneously treated with Platensimycin (PTM, 20 μM; Cayman Chemical, #15507) for 48 h. Macrophages were stained in Oil Red O working solution (3:2 dilution with distilled water of a stock solution of 0.5 mg/μl in 100% isopropanol) and hematoxylin and eosin (H&E). Stained coverslips were imaged using a Leica Thunder DM6 B Microsystems inverted fluorescent microscope. Areas positively stained with Oil Red O were quantified using the ImageJ color threshold toolkit. The intracellular lipid droplet area stained with Oil Red O was expressed as lipid droplet staining intensity relative to the total cell number.

Following conditioned media treatments, macrophages were lysed with 3 cycles of the standard freeze-thaw method in PBS, and lysates were normalized to protein concentration using the Bradford Protein Assay. Acetyl-CoA and FAS concentrations were determined using commercial ELISA kits according to the manufacturer's instructions (MyBioSource, #MBS9309791; Aviva Systems Biology, OKEH01027, respectively).

### Animal regulations and ethics

All animal protocols were approved by the Washington University in St. Louis institutional animal care and use committee (IACUC). Mouse housing, breeding, and experimental procedures were conducted in accordance with national and institutional guidelines and ethics. All animals were humanely euthanized in accordance with the guidelines set forth by the IACUC and the American Veterinary Medical Association (AVMA) Guidelines for the Euthanasia of Animals. Euthanasia was performed by trained personnel to minimize pain and distress and involved cervical dislocation followed by exsanguination and/or decapitation.

### Mouse models

Conditional liver-specific knockdown of *Fasn* on a C57BL6 background was achieved using previously reported floxed *Fasn*[fl/fl] mice that also express an albumin-*Cre* promoter (*Cre*[+])[56]. Liver *Fasn*[fl/fl] *Cre*[+] mice were crossed with *Apoe*[−/−] knockout mice (Jackson lab, strain #002052) to yield liver *Fasn*[fl/fl] *Cre*[+]*Apoe*[−/−] mice. At 7 weeks of age, male *Fasn*[fl/fl] *Cre*[+]*Apoe*[−/−] mice and male *Fasn*[+/+]*Cre*[−] *Apoe*[−/−] littermates were maintained on a continuous 42% high-fat diet for 16 weeks (Inotiv, TD.88137). On a weekly basis, body weights were recorded, and blood serum samples were collected from the tail vein. Similarly, male *Fasn*[+/+]*Cre*[−]*Apoe*[−/−] littermates were maintained on a 42% high-fat diet with PTM (100 mg/kg/day infused into the diet) for a 16-week treatment period. After 16 weeks, mice were sacrificed, and serum, hearts, aorta, liver, and white adipose tissue were collected for immediate analysis, embedded in OCT, or stored at −80 °C for later use.

### FAS enzyme activity and content assay

FAS enzyme activity was measured as previously described with modifications[21,22,69]. Liver and white adipose tissue were digested in a mammalian cell lysis kit (Millipore Sigma, MCL-1KT), and homogenates were centrifuged at 5000 g for 5 min at 4 °C. The supernatant and serum were standardized to 30 μg of total protein and added to 80 μL of assay buffer (1 M KPO4 buffer [pH 7], 50 mM EDTA [pH 8.0], 50 mM DTT, 1.1 mM NADPH (Millipore Sigma, N1630), 1 mM acetyl-CoA (Millipore Sigma, A2056). The rate of NADPH oxidation was monitored by measuring absorbance at 340 nm at 37 °C for 30 min in the absence, and then in the presence of 10 μL of the rate-limiting substrate malonyl-CoA for 10 min (1 mM; Millipore Sigma, M4263). Data was analyzed by calculating the OD decrease after correcting for the nonspecific background rate obtained without malonyl CoA substrate. FAS enzyme was defined as μmoles NADPH consumed·min$^{-1}$·mg$^{-1}$. An extinction coefficient of 6220 M$^{-1}$ cm$^{-1}$ was used in the specific activity calculation utilizing Beer's law as previously described[69].

### Aortic atherosclerotic burden assessment

Murine hearts were harvested *en bloc* at the time of sacrifice after 16 weeks of diet treatment. The tissue was fixed in OCT compound (Fisher Scientific), and the aortic valve region was sectioned at 10 μm thickness. Sections were then fixed in 4% paraformaldehyde (PFA), followed by 60% isopropanol for 5 min. Aortic valve sections were then stained in an Oil Red O working solution (3:2 dilution with distilled water from a stock solution of 0.5 mg/μl in 100% isopropanol). Valve plaque area was quantified in a blinded fashion using ImageJ as a percentage of plaque area in the aortic lumen, as previously described[70]. Corresponding sections of aortic value were also stained with 1:50 mouse anti-CD68 antibody (Bio-Rad, MCA1957). The primary antibody was detected with a 1:100 secondary antibody, donkey anti-rat IgG labeled with Alexa Fluor® 555 (Thermo Fisher Scientific, A78945), followed by a DAPI nucleus stain. Stained sections were then imaged on Leica Thunder DM6 B Microsystems inverted fluorescent microscope. The percentage of positively stained CD68 area relative to the total aortic lumen was quantified using ImageJ software.

Similarly, the entire aorta from the aortic arch to the infrarenal aortic bifurcation was microdissected and resected *en bloc* at the time of sacrifice. Harvested aortic specimens were fixed in 4% PFA for 24 h. The tissue was then effaced and stained using Oil Red O and imaged with Lecia S9i Microsystem microscope The area of plaque that positively stained with Oil Red O was taken relative to the aortic valve segment area using ImageJ in a blinded fashion[71].

### Immunofluorescence FASN staining

Differentiated macrophages were seeded onto fibronectin-coated coverslips in 24-well plates at a density of 240,000 cells per well. The cells were treated with either PTM (20 μM) or DMSO under various human serum conditions: high cFAS/low LDL, low cFAS/low LDL, low cFAS/medium LDL, and low cFAS/low LDL for 48 h. After treatment, the cells were rinsed three times with PBS and fixed with 4% PFA for 10 min at room temperature. Following fixation, the cells were washed with PBS and permeabilized with a 0.5% Triton X-100 solution for 15 min at room temperature. The cells were then blocked with 5% goat serum albumin for 1 h at room temperature,

**Table 1 | Human serum lipid profile**

| Sex | Groups | Condition | Age | LDL(mg/dL) | FAS(ng/ml) |
|-----|--------|-----------|-----|------------|------------|
| Male | FAS | High (n = 3) | 69.3 ± 5.85 | 63.0 ± 20.5 | 21.7 ± 4.41 |
| | | Medium (n = 3) | 62.3 ± 18.4 | 79.6 ± 28.5 | 5.45 ± 0.12 |
| | | Low (n = 3) | 62.6 ± 6.50 | 93.0 ± 25.8 | 0 |
| | LDL | High (n = 3) | 51.3 ± 19.3 | 183 ± 7.57 | 0 |
| | | Medium (n = 3) | 66.6 ± 8.32 | 103 ± 1.15 | 0 |
| | | Low (n = 3) | 69.3 ± 8.73 | 35.6 ± 10.1 | 0 |

Human serum demographics and lipid concentration from institutional serum biobank (n = 9) that were utilized for foam cell lipid formation experiments.
*LDL* low-density lipoprotein, *cFAS* circulating fatty acid synthase, ±SEM.

followed by incubation with FAS-Alexa®488 antibody (Santa Cruz, SC-48357; 1:100 dilution in PBS with 1% goat serum) for 18 h at 4 °C. The next day, the cells were washed three times with PBS for 5 min at room temperature. Coverslips were mounted on slides using DAPI-containing mounting media. Imaging was performed using a Leica Thunder DM6 B Microsystems inverted fluorescent microscope, and analysis was conducted with ImageJ software.

### Tissue histology

Murine liver and white adipose tissue were immediately harvested at the time of sacrifice and were embedded in OCT. Tissue was sectioned at 10 μm thickness and fixed in 4% PFA and stained by H&E. Sections were also immunostained with 2% donkey blocking agent for 1 h at room temperature then incubated with 1:100 primary mouse monoclonal FAS antibody (Santa Cruz, SC-48357), or 1:50 mouse anti-CD68 antibody (Bio-Rad, MCA1957). The primary antibody was detected with a 1:100 secondary antibody donkey anti-rat IgG labeled with Alexa Fluor® 555 (Thermo Fisher Scientific, A78945), followed by a DAPI nucleus stain. Imaging assessments were performed using a Leica Thunder DM6 B Microsystems inverted fluorescent microscope, and staining was quantified using the ImageJ software integrated density toolkit. The results were reported using consistent arbitrary units (a.u.)[72].

### Statistical analysis

All raw data files are provided in Supplementary Data 1. Statistical correlations between continuous variables such as distinct samples of serum cFAS, tissue FAS, content, or activity were evaluated using linear regression. Non-parametric two-tailed Mann–Whitney tests were used to assess the differences between inter- and intra-group analysis. Endpoints obtained over a time course were evaluated using two-way ANOVA with multiple comparisons. All analyses were performed on distinct samples using GraphPad Prism (Prism 9.1 software, GraphPad Software Inc.). We considered $p < 0.05$ statistically significant, $*p < 0.05$, $**p < 0.01$, $***p < 0.001$; ns non-significant. All graphical data are presented as mean ± SEM.

### Reporting summary

Further information on research design is available in the Nature Portfolio Reporting Summary linked to this article.

### Data availability

Primary raw data are provided in Supplementary Data 1. Additional requests for data and methods are available upon reasonable request to the corresponding author.

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

## Acknowledgements

This work was supported by grants from the Washington University School of Medicine Diabetes Research Center NIH/NIDDK P30 DK020589, NIH/NHLBI R01 HL153262 (M.A.Z.), NIH/NHLBI R01 HL150891 (M.A.Z.), NIH/NIDDK R01 DK101392 (C.F.S.), and NIH/NHLBI R01 HL157154 (C.F.S.). The Washington University School of Medicine Mass Spectrometry Facility is supported by US Public Health Service Grants P41 GM103422 and P60 DK20579.

## Author contributions

Conception & Design: R.M., D.I., M.Z., M.X.G.I., C.F.S., M.A.Z. Analysis & Interpretation: R.M., D.I., C.E., L.B., B.A., F.F.H., S.A., R.C., M.Z., M.X.G.I., C.F.S., M.A.Z. Data Collection: R.M., D.I., C.E., L.B., B.A., F.F.H., S.A., R.C. Writing the Manuscript: R.M., D.I., M.Z., M.X.G.I., C.F.S., M.A.Z. Critical Revision: R.M., D.I., B.A., S.A., M.Z., M.X.G.I., C.F.S., M.A.Z. Approval of the Manuscript: R.M., D.I., C.E., L.B., B.A., F.F.H., S.A., R.C., M.Z., M.X.G.I., C.F.S., M.A.Z. Aggregable to be Accountable: R.M., D.I., C.E., L.B., B.A., F.F.H., S.A., R.C., M.Z., M.X.G.I., C.F.S., M.A.Z. Statistical Analysis: R.M., D.I., C.E. Obtaining Funding: C.F.S. and M.A.Z.

## Competing interests

M.Z. is co-founder of AirSeal CardioVascular, Inc., a biomedical startup company that aims to clinically translate diagnostic approaches for individuals suffering from complications related to atherosclerotic cardiovascular disease. All other authors declare no competing interests.
