## [Transparent Peer Review file · Communications Biology]

Targeting Fatty Acid Synthase Reduces Aortic Atherosclerosis and Inflammation

Corresponding Author: Dr Mohamed zayed

This manuscript has been previously reviewed at another journal. This document only contains information relating to versions considered at Communications Biology.

Version 0:

Reviewer comments:

Reviewer #1

(Remarks to the Author)

This is an interesting work that explores the role of fatty acid synthase (FAS) in atherosclerosis. However, there are several aspects that need to be addressed to draw more accurate conclusions. Specifically, clarity is needed regarding the form of FAS, the cell types involved, and the mechanism by which FAS depletion in the liver reduces atherosclerosis burden.

Specific Comments:

LDL as the Carrier for cFAS:

How do the authors ensure that cFAS-containing serum and LDL-containing serum are not mixed? This distinction is crucial for interpreting the results correctly.

Correction Needed:

The statement "PTM only reduced intracellular macrophage FAS activity in cells that were conditioned with high cFAS" is incorrect. Figure 1H shows that FAS activity is not increased in PTM-treated macrophages. Please correct this and provide a possible mechanism for the observed results.

Intracellular FAS Staining:

It is important to stain for intracellular FAS in the samples to support the conclusions drawn.

Adipose Tissue FA Increase:

How do the authors explain the significant increase in adipose tissue fatty acids in PTM-treated Apoe^{-/-} animals?

Sample Size in Animal Experiments:

In several experiments, the number of animals is too small. Animal studies should include a minimum of 6-8 animals per group to ensure statistical significance.

Comprehensive Lipid Profile Characterization:

A more comprehensive characterization of the lipid profile is required, including what happens with reverse cholesterol transport and HDL levels.

Correction of Figure Numbers:

The last section contains incorrect figure references. It should refer to Figure 6.

Adipose Tissue FAS Expression:

The change in expression of FAS in the adipose tissue of PTM-treated animals is very significant (Figures 3G and 6E). It is not modest as mentioned in the text. Please correct this.

Rationale for Studying Adipose Tissue:

There is no clear rationale or conclusion regarding the study of FAS expression and lipid metabolism in adipose tissue. Please provide a detailed explanation.

Animal Nomenclature Consistency:

Clarify if all animals used were floxed, and use consistent nomenclature throughout the manuscript.

Evidence for cFAS Participation:

There is no clear evidence for the participation of cFAS in this mechanism. Similarly, the role of intracellular FAS in the cells involved in the atherosclerotic plaque formation is not explored.

Relevant Literature:

Modulation of VSMC to foam cell during atherosclerosis is an important mechanism of the disease (10.1161/CIRCULATIONAHA.113.005015 & 10.1161/CIRCRESAHA.115.304634). A recent paper describes a critical role for FASN in the VSMC to foam cell transition (10.3390/ijms23031308). Similarly, FASN's role in macrophage (myeloid cell lineage)-derived foam cells is reported as critical (10.1074/jbc.M110.100321). Both reports highlight increased cholesterol efflux as a critical mechanism for inhibiting foam cell formation in Fas-deficient cells. Co-staining of SMA and cd68 with FAS, SOAT1, and ABCA1 in aortic tissue would be critical for correct data interpretation in this manuscript. More in-depth discussion of these two reports would be important.

Minor Comments:

In the introduction, after "30-60%," please add a period and remove the comma.

Reference 17 (review) does not match the statement, "Similarly, while de novo synthesis of fatty acids is integral to lipid homeostasis, the process is essential for orchestrating the transformation of monocytes into foam cells, which play a fundamental role in perpetuating atheroprogession and exacerbating plaque instability and vulnerability." Please find the precise research article to support this statement.

Reviewer #2

(Remarks to the Author)

The authors report the results of their exploration of the hypothesis that circulating fatty acid synthase (cFAS) may promote macrophage foam cell formation and atherosclerosis. The study employs in vitro models of foam cell formation using U937 cells, and a hepatocyte-specific knockout of FASN on the ApoE KO background to investigate the hypothesis. The authors conclude that hepatic expression of FAS is a key contributor to atherosclerosis in ApoE KO mice, and that the enzymatic activity of this enzyme in the extracellular space is a key regulator of foam cell formation in macrophages. However, readers may find it difficult to agree with these conclusions, given several issues with the present manuscript that the authors may find helpful to address.

Major points

- 1) The concentrations of cFAS reported to be present in the serum samples reach extremely high levels (" $>17\mu\text{g}/\mu\text{L}$ ", equivalent to 17 grammes per litre!). Presumably this is a typo?
- 2) The title of Figure 1 is misleading: "Fig 1. cFAS induces macrophage foam cell formation in vitro", since the results shown here are mainly from a small correlation analysis ($n=6$ or 8 samples), and association does not prove causality. Fig 1G does not prove this as the PTM could easily inhibit the endogenous, cytosolic FASN within the macrophages, or perhaps other cellular pathways, which has not been explored. This is a relevant point here, as it can be seen that the difference in FAS activity between high cFAS and low cFAS samples, or those treated or without PTM, show only a small change in enzymatic activity. A more suitable way to test the hypothesis in this system would be to challenge the macrophages with recombinant FAS protein and a control non-FAS recombinant protein.
- 3) In Figure 1B - is the labelling back to front? Visually it looks like there is higher foam cell formation in the low cFAS image.
- 4) Figure 2C shows total cholesterol - why not also check ApoB and non ApoB cholesterol? Likewise, there is reporting of liver and adipose triglyceride content, but why no mention of serum triglyceride content? This seems highly relevant in the context of the present study.
- 5) In Figure 3A, it is not clear why FAS inhibition might lead to less release of cFAS from liver cells – could the authors offer a possible mechanism? Why would hepatic FAS activity result in hepatocyte membrane damage or leakage?
- 6) Figure 3H shows a complete ablation of FAS activity in adipocytes after treatment of mice with PTM for 16 weeks. If PTM is so effective, would this not result in similar lack of FAS activity systemically? This is very difficult to believe, in light of the previously published work showing that global FASN deficiency is lethal (in fact this is the rationale for doing conditional knockouts for this gene [reference 50]).
- 7) There is an assumption that the biological effects seen are due to the enzymatic activity of cFAS. However, there is no mention of what the concentrations of the enzyme substrates acetyl-CoA, malonyl-CoA and NADPH may be in serum, and how this may affect the ability of the enzyme to function. It is very unlikely that they are present at sufficient concentrations to allow functional synthesis of fatty acids by cFAS. The rationale for using PTM to inhibit cFAS activity is therefore unclear. Could cFAS be a bridging molecule linking LDL to, for example, scavenger receptors instead?
- 8) The results clearly show less plaque in the presence of PTM, but it is premature to conclude that this is due to FAS inhibition, since there could be many other pathways affected by this small molecule. Figures 6C and 6F are particularly relevant here, as PTM clearly has a very pronounced anti-inflammatory effect in liver and adipose tissue. Could this not reflect a general anti-inflammatory effect systemically in other tissues, which would obviously have a large impact on atherogenesis?
- 9) The liver FAS knockout model offers a more robust testing of the hypothesis than the PTM experiments. However, a

plausible mechanism should be put forward to explain how the absence of hepatocyte FAS might affect atherogenesis. FAS is a large enzyme that would not escape the cytosol without substantial cellular damage. Could circulating levels of FAS simply reflect damage to hepatocytes (well established to occur in response to HFD in mice), in a fashion similar to ALT or other cytosolic markers?

10) The proteome of LDL-associated proteins is highly diverse, with dozens of proteins attaching to it loosely in the circulation. It may be helpful to mention this in the discussion. Perhaps cFAS levels are merely a marker or correlate of another protein which may be responsible for the observed effects?

Minor points

- 1) Table 1 indicates that there were 6 plasma samples available for study, but Fig 1 C and E indicates that 8 were used for the foam cell formation correlation analysis. Which is correct?
- 2) PTM is a potent antibiotic, so will likely have significant effects on the microbiota profile, a property that itself is linked to liver disease and atherosclerosis. It may be helpful to mention this possibility in the discussion.
- 3) It is surprising that there is not more discussion of ref [27], which also reports the effects of hepatic FASN knockout on metabolism.
- 4) It would also help to mention the study of O'Farrell et al Sci Rep 2022, as they report that FASN inhibition results in anti-inflammatory effects in liver. As it is well established that hepatic inflammation can drive atherosclerosis, this effect could be a key driver of the observed results.

Reviewer #3

(Remarks to the Author)

This article addresses the role of Fatty Acid Synthase (FAS) in atherosclerosis progression. They show that circulating FAS levels may promote the formation of foam cells in vitro, and regulates the formation of atherosclerotic plaque formation. Macrophages treated with human serum containing circulating FAS enhanced foam cell formation in vitro. The authors used Platensimycin (PTM) to inhibit circulating FAS levels, which was also associated with attenuated foam cell formation, and treatment of Apoe^{-/-} mice with PTM reduced atherosclerosis progression in a HFD model. Conditional deletion of FAS in liver also replicated the reduction in atherosclerotic plaque progression, which was further supported by the lack of macrophage accumulation in the artery.

Overall, the topic and data are of interest to the cardiovascular field. However, several limitations of the study exist and decrease enthusiasm for the manuscript. Several of these limitations are outlined below. If addressed, these would dramatically strengthen the rigor and impact of the study.

1. The molecular mechanism of how FAS-deletion in the liver affects atherosclerotic plaque formation is not described or a reasonable connection is not laid out in the study. In addition, basic mechanisms regulating atherosclerosis progression are not studied in the in vivo model which might help the reader to understand the connection between FAS-deletion and strong protection of atherosclerosis. Some examples might include LDL accumulation in the intima, endothelial activation/damage, monocyte recruitment, and foam cell formation/survival. In addition, earlier time points of disease progression (6-8 weeks?) may be justified given the major difference observed in lesion size at 16 weeks HFD.
2. The cholesterol report is interesting, but it may be more relevant to understand the ratios of vLDL, LDL, and HDL in the control and FAS-inhibited animals. Shifts in these balances may be key to understanding drivers of atherosclerosis progression.
3. Furthermore, it is well known that LDL alone does not promote foam cell formation in vitro. It may be interesting to compare cFAS-induced foam cells with more traditional approaches such as modified LDL (oxLDL or AcLDL), or also in combination. These studies might help the reader understand how cFAS modifies established foam cell assays, and potentially synergizes with them.
4. It's unclear why macrophages are reduced in the liver of the Liver-specific Fas-flox mouse model. Is this also true prior to HFD feeding and in other tissues? If there is a systemic macrophage defect in this mouse, it might help understand the reduced baseline weight and why they are so resistant to atherosclerosis formation.

Several technical concerns existed:

1. How are foam cells determined in the in vitro assessment in Figure 1? Is there a quantitative definition for this assay, or is it a qualitative approach? Images need scale bars.
2. Individual replicates need to be shown in all figures. Moving away from scale bars will allow for a better assessment of the data quality and rigor.
3. What sex were animals used in the in vivo studies? Could the authors please provide separate analysis for the different sexes?
4. Could the authors report the number of independent times experiments were performed, and the number of replicates used in each study?
5. Plaque analysis in the Aortic sinus is unconventional; this site is typically measured as Plaque Area- using a size measurement, instead of a %.
6. What are the AU that CD68 is measured in? Typically this would be an area measurement or a percentage of CD68+/Plaque area.
7. Figure 6 is not referenced appropriately in the text.
8. Financial disclosures are not clearly stated.

Version 1:

Reviewer comments:

Reviewer #1

(Remarks to the Author)

The authors have adequately addressed my previous concerns. I believe this paper presents novel information that will be of significant interest to the readers.

Reviewer #2

(Remarks to the Author)

No further comments.

Reviewer #3

(Remarks to the Author)

This is an interesting study. The revised paper addressed any technical concerns that I had regarding the data/rigor of the study. It would be nice to have additional mechanisms, but that is likely outside the scope of this manuscript and I agree with the authors on this point.

Response to Reviewers:

Reviewer #1:

This is an interesting work that explores the role of fatty acid synthase (FAS) in atherosclerosis. However, there are several aspects that need to be addressed to draw more accurate conclusions. Specifically, clarity is needed regarding the form of FAS, the cell types involved, and the mechanism by which FAS depletion in the liver reduces atherosclerosis burden.

1. Form of FAS:

In our study we focused on cytoplasmic FAS. To elucidate this, we include in the revised manuscript immunofluorescence (IF) staining under various serum cFAS/LDL conditions, and with and without PTM treatment. Our results, as shown in **Figure 2**, demonstrate an increase in cytoplasmic FAS specifically under high cFAS/low LDL conditions, while other conditions did not show as much of an impact (**page 9, lines 1-13; page 12, lines 1-14**).

2. FAS content versus activity:

Importantly, our data indicate that PTM treatment does not lead to FAS degradation or reduction of its intracellular protein levels. Instead, we observed that PTM impacts FAS activity, as demonstrated in new data now presented in **Figures 1G and 1H**. Additionally, PTM treatment reduces the formation of lipid droplets in macrophages, further highlighting its effect. This distinction between FAS content and activity is crucial in our understanding of the mechanism of action of FAS and its inhibition by selective agents such as PTM (**page 11, lines 1-16**).

3. Functional consequences:

We observed that reduction in FAS activity resulted in decreased lipid droplet accumulation in macrophages. We now show this using multiple FAS inhibitors including PTM, TVB-2640, and GSK2194069. We include these new data in **Figures 1I and J** (**page 11, lines 18-23**).

The statement “PTM only reduced intracellular macrophage FAS activity in cells that were conditioned with high cFAS” is incorrect. Figure 1H shows that FAS activity is not increased in PTM-treated macrophages. Please correct this and provide a possible mechanism for the observed results.

We appreciate the reviewer’s comments. **Figure 1** has been completely revised and experiments have been added to both **Figure 1** as well as the new **Figure 2** to address the reviewer’s comments.

We now clarify our interpretation of the results and the new **Figure 1J**. We now state:

“Treatment with the FAS inhibitor PTM (20µM) significantly decreased macrophage-derived foam cell formation when cells were conditioned with serum containing high cFAS (>16ng/mL) and low LDL (<90 mg/dL; **Fig. 1G**; p<0.01). Furthermore, intracellular FAS activity in macrophages that were conditioned with serum containing high cFAS was higher than macrophages conditioned with serum containing high LDL (**Fig. 1H**; p<0.05). Treatment with PTM resulted in a marked reduction of intracellular FAS activity in treated macrophages (**Fig. 1H**; p≤0.01), suggesting that cFAS is a potent factor in foam cell induction.” (**page 11, lines 12-18**)

To further clarify the potential cellular mechanism that led to the observations, we now provide additional significant data presented in the new **Figures 1I and J**. This shows that relative to untreated control, the use of multiple inhibitors leads to decreased macrophage foam cell formation (**page 11, lines 18-23**).

Furthermore, to distinguish between the impact of exogenous and endogenous FAS, we evaluated the intensity of FAS content in macrophages following treated with either high or low cFAS human serum. As reported in the new **Figure 2**, we observed that macrophages exposed to high cFAS/low LDL serum exhibited markedly elevated FAS immunostaining intensity compared to low cFAS serum conditions (**Figure 2A & B**). This was also corroborated using ELISA (**Figure 2C**). These findings suggest that cFAS is internalized by macrophages and contributes to foam cell formation, while PTM modulates FAS activity rather than intracellular content level (**page 12, lines 2-14**).

In the discussion we now also add additional insight about the potential cellular mechanism of action (**page 15, lines 17-20; page 18, lines 13-23; page 19, lines 1-5**).

Intracellular FAS Staining:

It is important to stain for intracellular FAS in the samples to support the conclusions drawn.

We appreciate this recommendation and have conducted new experiments that are presented in the new **Figure 2**. Here we show in **Figure 2A** immunofluorescent staining of intracellular FAS in macrophages following treatment with human serum with variable cFAS content, and with or without PTM. **Figure 2B** provides a quantitative summary of FAS immunofluorescent staining (**page 12, lines 2-14**).

Adipose Tissue FA Increase:

How do the authors explain the significant increase in adipose tissue fatty acids in PTM-treated *Apoe*^{-/-} animals?

We appreciate the reviewer's comment, as we were also intrigued by this finding. There are a few potential explanations for the observed increase in adipose tissue fatty acids in PTM-treated *Apoe*^{-/-} animals, despite their reduced FAS activity and content (**Figure 3I-J**). Here are 3 potential mechanisms that can explain this observation.

1. First, it is possible that this reflects a compensatory fatty acid uptake. With reduced local FAS activity, adipocytes may be compensating for this by increasing fatty acid uptake from the circulating pool to maintain adequate lipid stores.
2. Second, there may be altered lipid metabolism in *Apoe*^{-/-} animals that have reduced FAS content/activity due to selective genetic knockdown or treatment with PTM. As a result, there may be a shift in lipid handling towards increased storage of circulating fatty acids.
3. Third, there may be reduced fatty acid oxidation. With lower FAS activity this may lead to decreased malonyl-CoA, a known inhibitor of fatty acid oxidation, potentially resulting in reduced fatty acid breakdown and resultant higher levels in the adipose tissue.

We now provide an overview of these potential mechanisms in the manuscript discussion (page 19, lines 22-23; page 20, line 1-5).

Sample Size in Animal Experiments:

In several experiments, the number of animals is too small. Animal studies should include a minimum of 6-8 animals per group to ensure statistical significance.

We appreciate the reviewer's concern about sample sizes. We have now increased the sample size for all groups to n=6 for data presented in Figure 3 (see Figure 3 legend; page 23, lines 9-17). The samples size for longitudinal serum measurements have also been increased in Figure 4 (page 23, lines 19-22; page 24, lines 1-3). Additionally, the sample size for aortic plaque assessments has been increased to n=8-17 (page 24, lines 5-10).

Comprehensive Lipid Profile Characterization:

A more comprehensive characterization of the lipid profile is required, including what happens with reverse cholesterol transport and HDL levels.

In Figure 4 (E-H) we show that PTM treatment did not significantly alter serum cholesterol, triglycerides, free fatty acids, or glucose levels, suggesting a stable metabolic state. However, a notable reduction in FAS content and activity at weeks 9 and 16 points to a potential suppression of lipogenesis (Figures 4 A-D), which could lead to reduced lipid accumulation in aorta and improved metabolic health over time. This finding is particularly relevant given the role of FAS in lipid synthesis and metabolism.

We agree with the reviewer that future studies will be needed to further explore HDL, reverse cholesterol transport (RCT) markers, and their long-term effects on lipid metabolism and atheroprogession. Unfortunately, in our current study and murine cohort we were highly constrained by the serum volume that we can extract from the mice at any given time point (even at the time of sac). There was limited feasibility to objectively obtain HDL content in addition to total cholesterol, FFA, TG, glucose, and FAS content. We estimate that at least 10 additional mice per cohort would be needed to provide an adequate pooled serum sample volume for standard FLPC techniques to measure HDL content (as we previously described; De Silva et al. Atherosclerosis 2019; 287: 38-45). It is indeed our plan to conduct these studies in the future.

Correction of Figure Numbers:

The last section contains incorrect figure references. It should refer to Figure 6.

This has now been corrected for a new Figure 7.

Adipose Tissue FAS Expression:

The change in expression of FAS in the adipose tissue of PTM-treated animals is very significant (Figures 3G and 6E). It is not modest as mentioned in the text. Please correct this.

We appreciate the reviewer's comment. We now make this correction our description of new Figure 3I (page 13, lines 12-13) and Figure 7F (page 15, lines 4-14).

Rationale for Studying Adipose Tissue:

There is no clear rationale or conclusion regarding the study of FAS expression and lipid metabolism in adipose tissue. Please provide a detailed explanation.

We analyzed both liver and white adipose tissue for two main reasons. First, *de novo* lipogenesis occurs largely in the liver and white adipose tissue. Therefore, we hypothesized that genetic or pharmacological targeting of FAS was going to likely disrupt FAS content in either or both of these tissues. Second, we intended to add robustness to our study. Rather than just analyze the impact of FAS targeting in the liver, we aimed to determine if our observed phenotype in the serum and aorta is impacted by multiple tissue types.

We observed that indeed both the liver and adipose tissue are likely playing an important balance in FAS tissue expression. Specifically, in **Figure 3H**, in *Fasn^{fl/fl} Cre⁺ Apoe^{-/-}* mice we observed a significant reduction in FAS content in the liver tissue (as expected). Interestingly, , we observed the opposite effect in white adipose tissue (**Figure 3I**). Here we observed that targeted knockdown of *Fasn* in liver tissue caused a significant increase in FAS content in the white adipose tissue. This clearly demonstrates a likely compensatory mechanism that exists when FAS content in the liver is reduced and may help explain why serum free fatty acids (FFAs) are not significantly different in *Fasn^{fl/fl} Cre⁺ Apoe^{-/-}* mice (**Figure 4G**). We now highlight these findings in the results (page 13, lines 3-15) an explanation of this is also now provided in the discussion (page 19, lines 18-23; page 20, lines 1-5).

Animal Nomenclature Consistency:

Clarify if all animals used were floxed and use consistent nomenclature throughout the manuscript.

We now clarify that “conditional liver-specific knockdown of *Fasn* was achieved using previously reported floxed *Fasn^{fl/fl}* mice that also express an albumin-Cre promoter (*Cre⁺*)” (page 7, lines 2-3).

Evidence for cFAS Participation:

There is no clear evidence for the participation of cFAS in this mechanism. Similarly, the role of intracellular FAS in the cells involved in the atherosclerotic plaque formation is not explored.

We appreciate this critique and have conducted additional experiments to alleviate this concern. These experiments are now presented in **Figures 1 & 2**.

In **Figure 1 I and J**, we show how pharmacological targeting of cFAS in human serum with multiple pharmacological agents (PTM, TVB-2640, and GSK2194069) all lead to the same reduction in macrophage foam cell formation. These experiments show that inhibition of cFAS in the serum has direct impact on foam cell formation.

In **Figure 2**, we show that treatment of macrophages with human serum containing high cFAS leads to increased intracellular FAS content (in both conditions that were treated and untreated

with PTM). These experiments show that intracellular FAS is impacted by levels of extracellular cFAS.

These results are described in the revised manuscript (page 11, lines 18-23; page 12, lines 2-12).

Minor Comments:

In the introduction, after "30-60%," please add a period and remove the comma.

Thank you for the edit which has been made (page 3, line 7).

Reference 17 (review) does not match the statement, "Similarly, while de novo synthesis of fatty acids is integral to lipid homeostasis, the process is essential for orchestrating the transformation of monocytes into foam cells, which play a fundamental role in perpetuating atheroprogession and exacerbating plaque instability and vulnerability." Please find the precise research article to support this statement.

We appreciate the correction. The correct reference is now included as the new reference #17 (page 3, lines 19-22).

Reviewer #2:

The authors report the results of their exploration of the hypothesis that circulating fatty acid synthase (cFAS) may promote macrophage foam cell formation and atherosclerosis. The study employs in vitro models of foam cell formation using U937 cells, and a hepatocyte-specific knockout of FASN on the ApoE KO background to investigate the hypothesis. The authors conclude that hepatic expression of FAS is a key contributor to atherosclerosis in ApoE KO mice, and that the enzymatic activity of this enzyme in the extracellular space is a key regulator of foam cell formation in macrophages. However, readers may find it difficult to agree with these conclusions, given several issues with the present manuscript that the authors may find helpful to address.

Major points

1) The concentrations of cFAS reported to be present in the serum samples reach extremely high levels (">17ug/uL", equivalent to 17 grammes per litre!). Presumably this is a typo?

We appreciate the reviewer pointing our attention to this error. We have now corrected this typo. The intended units are ng/μl. This is now reflected in the text and new figures.

2) The title of Figure 1 is misleading: "Fig 1. cFAS induces macrophage foam cell formation in vitro", since the results shown here are mainly from a small correlation analysis (n=6 or 8 samples), and association does not prove causality. Fig 1G does not prove this as the PTM could easily inhibit the endogenous, cytosolic FASN within the macrophages, or perhaps other cellular pathways, which has not been explored. This is a

relevant point here, as it can be seen that the difference in FAS activity between high cFAS and low cFAS samples, or those treated or without PTM, show only a small change in enzymatic activity. A more suitable way to test the hypothesis in this system would be to challenge the macrophages with recombinant FAS protein and a control non-FAS recombinant protein.

The **Figure 1** title is now changed to “Differential impact of cFAS and LDL on macrophage foam cell formation *in vitro*” (**page 22, line 2**).

We have conducted additional experiments to support our hypothesis that cFAS is a major player in macrophage foam cell formation. In our new **Figure 1I-J** we now present additional data using other cFAS inhibitors that demonstrate that a similar phenotype of reduced macrophage foam cell formation (**page 11, lines 18-23**). **Figure 2** shows that intracellular FAS increases with the addition of serum containing high cFAS (in PTM treated and untreated conditions; **page 12, lines 2-12**). **Supplemental Figure 1** shows that intracellular macrophage lipid droplet accumulation is dose dependent on the amount of high cFAS human serum added (**page 11, lines 4-11**). Taken together, these data demonstrate that cFAS is playing a major role in the observed phenotype.

Nevertheless, we acknowledge in the manuscript that the mechanism of how cFAS is impacting macrophage foam cell formation is not yet fully elucidated and is certainly a topic for our future studies (**page 18, lines 13-23; page 19, lines 1-9**).

3) In Figure 1B - is the labelling back to front? Visually it looks like there is higher foam cell formation in the low cFAS image.

1. Our correlation analysis (**Figure 1C**) shows that higher cFAS levels are indeed linked to more lipid droplets in foam cells ($R^2 = 0.445$, $P = 0.049$).
2. **Figures 1D-F** provide additional data to support this observation and show that when we combine high cFAS with LDL, we see a lot more lipid buildup in macrophages.
3. Importantly, high cFAS/low LDL conditions led to significantly more lipid droplets and FAS activity compared to low cFAS/high LDL (**Figure 1F**).

These results, based on multiple experiments, consistently support our conclusion that high cFAS promotes macrophage foam cell formation (**page 11, lines 12-23**).

4) Figure 2C shows total cholesterol - why not also check ApoB and non ApoB cholesterol? Likewise, there is reporting of liver and adipose triglyceride content, but why no mention of serum triglyceride content? This seems highly relevant in the context of the present study.

This is an excellent point about the value of including ApoB, non-ApoB cholesterol, and serum triglycerides in our analysis. While our initial report was limited due to restricted serum sample availability, we are pleased to add new data in **Figure 4** that includes total cholesterol, FFA, TG, and glucose from serum over weeks: 0, 3, 6 and 14.

5) In Figure 3A, it is not clear why FAS inhibition might lead to less release of cFAS from

liver cells – could the authors offer a possible mechanism? Why would hepatic FAS activity result in hepatocyte membrane damage or leakage?

We appreciate the reviewer's insightful question regarding the mechanisms behind reduced cFAS release from liver following FAS inhibition (now data is shown in **Figure 4A**). Two mechanisms may be the reason to address this observation:

- 1- We show in **Figures 3G and 3H** that the inhibition of FAS by PTM reduces both enzymatic activity and content of FAS in hepatocytes at week 16. This suggests that the reduction of FAS activity and content may lead to the subsequent decrease in the shedding of liver-derived cFAS into the serum.
- 2- We suggest that PTM may not only inhibit FAS activity, but may also potentially impact the interaction between FAS and ApoB. This interaction has been considered as a mechanism that facilitates the release of cFAS from the liver and into the bloodstream (**page 17, lines 22-23; page 18, lines 1-6**). The altered FAS-ApoB interaction could lead to reduced incorporation of cFAS into lipoprotein cargo, and thereby decrease its secretion from the liver.

6) Figure 3H shows a complete ablation of FAS activity in adipocytes after treatment of mice with PTM for 16 weeks. If PTM is so effective, would this not result in similar lack of FAS activity systemically? This is very difficult to believe, in light of the previously published work showing that global FASN deficiency is lethal (in fact this is the rationale for doing conditional knockouts for this gene [reference 50]).

We thank the reviewer for their valuable comments on the observed reduction of FAS activity in adipose tissues at week 16, as presented in **Figure 3J** (updated figure). We have addressed this observation with the following explanations:

1. In **Figure 3I-J**, we demonstrate that PTM treatment leads to a significant reduction in FAS content and activity in adipose tissues at week 16. We propose that this adipose FAS activity reduction is due to the direct impact of PTM treatment and is likely the result of a feedback mechanism triggered by the increased levels of FFAs in adipose tissue, as shown in **Figure 3F**.
2. The elevated FFAs could signal a downregulation of FAS activity as part of a homeostatic response to limit further fatty acid synthesis, thereby preventing excessive lipid accumulation. This feedback mechanism may explain why the reduction in FAS activity was not fatal, as it allows for the maintenance of essential metabolic functions despite the inhibition of FAS (**page 19, lines 22-23; page 20, lines 1-3**).

7) There is an assumption that the biological effects seen are due to the enzymatic activity of cFAS. However, there is no mention of what the concentrations of the enzyme substrates acetyl-CoA, malonyl-CoA and NADPH may be in serum, and how this may affect the ability of the enzyme to function. It is very unlikely that they are present at sufficient concentrations to allow functional synthesis of fatty acids by cFAS. The rationale for using PTM to inhibit cFAS activity is therefore unclear. Could cFAS be a bridging molecule linking LDL to, for example, scavenger receptors instead?

We now address these concerns in the revised manuscript as follows:

1. Regarding the concentrations of enzyme substrates: We have now included the concentrations of acetyl-CoA, malonyl-CoA, and NADPH in serum on **page 7, lines 14-22** and **page 8, lines 1-3**.
2. Rationale for using PTM to inhibit cFAS activity:
Our approach to using PTM for inhibiting cFAS activity is based on established methods for assessing FAS activity in serum and tissue samples. Specifically, we follow the method we previously published and validated in our Nature Scientific Reports paper (<https://www.nature.com/articles/s41598-021-98479-7>). In this study, we now demonstrate the efficacy of using PTM to inhibit FASN activity in various biological contexts, including serum samples.
3. Alternative roles of cFAS:
In our study, we discuss that FAS could be released into the blood associated with LDL particles, but importantly, not all LDL particles carry FAS (**page 17, lines 21-23; page 18, lines 1-12**). Our data in new **Supplementary Figure 1** demonstrate that the level of serum cFAS can significantly impact foam cell formation. In this experiment we used a serial dilution of high FAS/low LDL serum with human healthy donor serum, suggesting a direct role for cFAS in this process independent of LDL levels (**page 11, lines 6-11**).

8) The results clearly show less plaque in the presence of PTM, but it is premature to conclude that this is due to FAS inhibition, since there could be many other pathways affected by this small molecule. Figures 6C and 6F are particularly relevant here, as PTM clearly has a very pronounced anti-inflammatory effect in liver and adipose tissue. Could this not reflect a general anti-inflammatory effect systemically in other tissues, which would obviously have a large impact on atherogenesis?

We appreciate the reviewer's insightful comment regarding the interpretation of our results with PTM. As highlighted, PTM exhibits pronounced anti-inflammatory effects in both liver and adipose tissue (**Figures 7C and 7E**), which raises an important point about its potential systemic impact on inflammation. This could indeed contribute to changes in atherogenesis beyond just FAS inhibition.

To address this concern, we added clarifications in the Discussion (**page 20, lines 4-7**) to suggest that the anti-atherosclerotic effects observed with PTM treatment may have in part resulted from a combination of direct FAS inhibition and broader anti-inflammatory effects.

Additionally, we have included data on other FAS inhibitors, such as TVB-2640 and GSK2164069, in new **Figures 1I and 1J**, which provide clear evidence that like PTM they can also reduce foam cell formation *in vitro*. This suggests that the impact on foam cells is not solely due to PTM, but can also be mimicked with other known FAS inhibitors (**page 11, lines 18-23**).

9) The liver FAS knockout model offers a more robust testing of the hypothesis than the PTM experiments. However, a plausible mechanism should be put forward to explain how the absence of hepatocyte FAS might affect atherogenesis. FAS is a large enzyme that would not escape the cytosol without substantial cellular damage. Could circulating levels of FAS simply reflect damage to hepatocytes (well established to occur in response to HFD in mice), in a fashion similar to ALT or other cytosolic markers?

We appreciate the reviewer's insightful comment.

1. Unlike whole body *Fasn* deficiency is embryologically lethal, the conditional liver *Fasn* knockout model (*Fasn^{fl/fl} Cre⁻*) is not lethal (**page 19, lines 10-11**), as demonstrated in previous studies (**Reference 27**).

2. We discuss in the manuscript that PTM was previously observed to reduce *de novo* fatty acid synthesis in *db/db* mice (**page 19, lines 18-19; Reference 69**). This is consistent with our findings in *Fasn^{+/+} Cre⁻ Apoe^{-/-}* mice that were treated with PTM showed normal weight gain suggesting acute non-lethal dosing.

3. We hypothesize that FAS is released from the liver into the bloodstream. This is supported by our observations in the *Fasn* conditional liver knockout mice that were maintained on a high fat diet. Interestingly, in these mice we observed hyperlipidemia, but reduced release of cFAS in serum, as shown in **Figure 4A and 4E**, making this model a good control for our PTM-treated mice.

4. A recent study, "Fatty Acid Synthase Inhibitor Platensimycin Intervenes the Development of Nonalcoholic Fatty Liver Disease in a Mouse Model" (<https://pmc.ncbi.nlm.nih.gov/articles/PMC8773228/>; **Reference 66**) also demonstrated that PTM reduces liver steatosis and intervenes in the progression of NAFLD, indicating a protective impact rather than direct damage to liver cells. The same study shows that PTM reduces the protein levels of fatty acid synthase in HepG2 cells, suggesting a targeted action that may not inherently compromise cell viability. This study also showed that FAS inhibition reduced inflammatory markers IL-1 β and reducing transaminitis (elevated serum ALT and AST; **page 20, lines 8-11**).

5. We agree with the reviewer that the shedding of cFAS from the liver may be a pathological process, that is reflective of liver damage. This would be consistent with prior findings cited from **Reference 66 and 71; page 20 lines 8-11**), but also with our observations in new Figure 7, which shows that liver specimens from *Fasn^{fl/fl} Cre⁻ Apoe^{-/-}* mice and *Fasn^{+/+} Cre⁻ Apoe^{-/-}* mice treated with PTM there was a significant reduction of inflammation (**page 15, lines 4-6; page 18, lines 21-23**).

10) The proteome of LDL-associated proteins is highly diverse, with dozens of proteins attaching to it loosely in the circulation. It may be helpful to mention this in the discussion. Perhaps cFAS levels are merely a marker or correlate of another protein which may be responsible for the observed effects?

We now acknowledge that the LDL-associated proteome is highly diverse (**page 18, lines 1-4**).

To strengthen our findings and address the possibility that cFAS levels might be a marker or correlate of another protein responsible for the observed effects, we now provide additional data using multiple different and well characterized FAS inhibitors (PTM, TVB-2640 and GSK2194069) in our study. As shown in **Figure 1I and J**, TVB-2640 and GSK2194069 reduced lipid droplets in foam cells to a similar extent as PTM. These new findings support a direct role for FAS in this process.

We recognize the need for further validation of our hypothesis. In future experiments, we plan to use a fully recombinant functional FAS to test its impact on the macrophage foam cell formation.

This approach will help further isolate the specific role of FAS from potential confounding factors associated with LDL-bound proteins.

Minor points

1) Table 1 indicates that there were 6 plasma samples available for study, but Fig 1 C and E indicates that 8 were used for the foam cell formation correlation analysis. Which is correct?

We appreciate the reviewer's attention to detail regarding the sample numbers. To clarify, we have incorporated new serum samples, bringing the total to n=9 as now shown in both Table 1 (page 21, lines 2-5; page 25, lines 6-8) and legend of Figures 1C and 1D (page 22, line 7). We have updated these figures and the corresponding text to reflect this change, ensuring consistency throughout the manuscript.

2) PTM is a potent antibiotic, so will likely have significant effects on the microbiota profile, a property that itself is linked to liver disease and atherosclerosis. It may be helpful to mention this possibility in the discussion.

We concur with this point. PTM, as a potent antibiotic, has a broad spectrum of activity, particularly against Gram-positive bacteria. This property suggests that PTM administration could significantly alter the gut microbiota composition. Given the established links between gut microbiota, liver disease, and atherosclerosis, these microbial changes might contribute to PTM's observed effects.

We acknowledge the impact of PTM on the microbiota is current unknown (page 19, lines 18-19).

3) It is surprising that there is not more discussion of ref [27], which also reports the effects of hepatic FASN knockout on metabolism.

We appreciate your valuable comment. We have incorporated this information into our discussion on page 19, lines 5-9.

4) It would also help to mention the study of O'Farrell et al Sci Rep 2022, as they report that FASN inhibition results in anti-inflammatory effects in liver. As it is well established that hepatic inflammation can drive atherosclerosis, this effect could be a key driver of the observed results.

We appreciate the reviewer's suggestion to include the study by O'Farrell et al. (Sci Rep 2022) in our discussion. We have added this reference to our manuscript and incorporated its findings into our discussion (page 20, lines 8-11).

Reviewer #3:

This article addresses the role of Fatty Acid Synthase (FAS) in atherosclerosis progression. They show that circulating FAS levels may promote the formation of foam

cells *in vitro*, and regulates the formation of atherosclerotic plaque formation. Macrophages treated with human serum containing circulating FAS enhanced foam cell formation *in vitro*. The authors used Platensimycin (PTM) to inhibit circulating FAS levels, which was also associated with attenuated foam cell formation, and treatment of Apoe^{-/-} mice with PTM reduced atherosclerosis progression in a HFD model. Conditional deletion of FAS in liver also replicated the reduction in atherosclerotic plaque progression, which was further supported by the lack of macrophage accumulation in the artery.

Overall, the topic and data are of interest to the cardiovascular field. However, several limitations of the study exist and decrease enthusiasm for the manuscript. Several of these limitations are outlined below. If addressed, these would dramatically strengthen the rigor and impact of the study.

1. The molecular mechanism of how FAS-deletion in the liver affects atherosclerotic plaque formation is not described or a reasonable connection is not laid out in the study. In addition, basic mechanisms regulating atherosclerosis progression are not studied in the *in vivo* model which might help the reader to understand the connection between FAS-deletion and strong protection of atherosclerosis. Some examples might include LDL accumulation in the intima, endothelial activation/damage, monocyte recruitment, and foam cell formation/survival. In addition, earlier time points of disease progression (6-8 weeks?) may be justified given the major difference observed in lesion size at 16 weeks HFD.

We appreciate the reviewer's insightful comments regarding the molecular mechanisms linking hepatic FAS deletion to atherosclerotic plaque formation. While our study demonstrates a strong protective effect of liver-specific FAS deletion on atherosclerosis progression, we acknowledge that further investigation into the underlying mechanisms is warranted.

1. We suggest that liver knockout of *Fasn* reduces the release of cFAS into the bloodstream, which in turn reduces atherosclerosis (page 17, lines 20-23; page 18, line 1). This is confirmed in Figure 4A, where PTM treatment reduces cFAS in serum but not in liver at week 16, as shown in Figure 3G. Our *in vitro* results in Figures 2A-C further support this, demonstrating that PTM treatment does not affect the content of FAS but reduces FAS activity in macrophages, as shown in Figures 1G-H. These findings align with the results reported by (Reference 69), who found that PTM treatment reduced liver FAS activity.
2. PTM treatment reduces lesion size and macrophage recruitment in the aortic valve, as shown in Figures 6A-C. Similar reductions in inflammation are observed in liver and adipose tissue, as demonstrated in Figures 7A, 7C, 7D, and 7G (page 15, lines 4-14). These results collectively support a systemic anti-inflammatory effect with PTM treatment, which may contribute to the observed reduction in atherosclerosis progression.
3. Regarding the relationship between endothelial cells and liver FAS knockout, a study by Chakravarthy et al. (Reference 27) that liver-specific FAS knockout mice had altered lipid profiles and metabolic adaptations. While this study did not directly examine endothelial function, it suggests that hepatic FAS deletion can have systemic effects on lipid metabolism, which could potentially impact endothelial cell function. Further research is needed to elucidate the specific mechanisms linking conditional liver *Fasn* knockdown, endothelial cell function, and atherosclerosis.
4. It is our experience and the experience of prior investigations that Apoe^{-/-} mice do not develop significant plaque burden until they are maintained on a high-fat diet for at least 14-16 weeks (Kong YY, Li GQ, Zhang WJ, Hua X, Zhou CC, Xu TY, Li ZY, Wang P, Miao CY.

Nicotinamide phosphoribosyltransferase aggravates inflammation and promotes atherosclerosis in ApoE knockout mice. *Acta Pharmacol Sin.* 2019 Sep;40(9):1184-1192. doi: 10.1038/s41401-018-0207-3. Epub 2019 Mar 4. PMID: 30833708; PMCID: PMC6786310.) This is why we selected 16 weeks for aorta and aortic valve harvest and analysis (**Figure 5**). On the other hand, we collected serum samples longitudinally at 3, 6, and 14 weeks to facilitate serum lipid analysis (**Figure 4**).

2. The cholesterol report is interesting, but it may be more relevant to understand the ratios of vLDL, LDL, and HDL in the control and FAS-inhibited animals. Shifts in these balances may be key to understanding drivers of atherosclerosis progression.

Figures 4A-D demonstrate a notable reduction in FAS content and activity at weeks 9 and 16 following PTM treatment, suggesting potential suppression of lipogenesis. This could lead to reduced lipid accumulation in the aorta and improved metabolic health over time (**page 13, lines 19-22; page 14, lines 1-2**). However, as shown in **Figures 4E-H**, PTM treatment did not significantly alter serum cholesterol, triglycerides, free fatty acids, or glucose levels, indicating metabolic stability (**page 14, lines 2-5**). These findings suggest that while PTM effectively reduces FAS activity, it maintains overall metabolic balance in the serum.

We recognize the importance of exploring HDL and RCT markers in future studies. Limited serum volume in our current cohort limited feasibility to obtain HDL measurements.

3. Furthermore, it is well known that LDL alone does not promote foam cell formation in vitro. It may be interesting to compare cFAS-induced foam cells with more traditional approaches such as modified LDL (oxLDL or AcLDL), or also in combination. These studies might help the reader understand how cFAS modifies established foam cell assays, and potentially synergizes with them.

In our study, we used different fatty acid synthase inhibitors (PTM, TVB-2640, GSk2194069) to confirm the reduction in foam cell formation when FAS is inhibited (new **Figure 1I-J**). This approach helped us further validate that the observed effect is specific due to FAS inhibition rather than an off-target effect of a single inhibitor (**page 11, lines 18-23; page 15, lines 19-20**).

Additionally, we performed experiments using a serial dilution of high cFAS/low LDL serum (**Supplementary Figure 1**). These demonstrated a dose-dependent decrease in foam cell formation as the percentage of high cFAS/low LDL serum was significantly reduced ($p=0.0137$). This gradient effect further supports the role of cFAS in foam cell formation and provides a nuanced understanding of the relationship between cFAS levels and foam cell development (**page 11, lines 8-11**).

4. It's unclear why macrophages are reduced in the liver of the Liver-specific Fas-flox mouse model. Is this also true prior to HFD feeding and in other tissues? IF there is a systemic macrophage defect in this mouse, it might help understand the reduced baseline weight and why they are so resistant to atherosclerosis formation.

The comment raises important questions about macrophage populations in *Fasn^{fl/fl} Cre⁺ ApoE^{-/-}* mice. We now cite references 66 and 71 that show that FAS inhibition is known to reduce inflammatory markers such as IL-1 β and reducing transaminitis (**page 20, lines 9-11**), and that

the *Fasn^{fl/fl} Cre⁺* (not on an *Apoe^{-/-}* background) has a normal phenotype with no known innate or cellular immune deficiencies (**Reference 27**).

We hypothesize that conditional *Fasn* in the liver likely leads to decreased availability of lipid substrates for inflammatory cytokine release in the liver (**Reference 66 and 71**). This anti-inflammatory effect may explain the observed reduction in macrophage numbers in mice that had either conditional knockdown of *Fasn* or pharmacological treatment with PTM (**page 20, lines 6-11**).

5. Several technical concerns existed:

1. How are foam cells determined in the in vitro assessment in Figure 1? Is there a quantitative definition for this assay, or is it a qualitative approach? images need scale bars.

We thank the reviewer for raising this important point about our foam cell quantification method. To address this concern, we have clarified our approach as follows. To quantify foam cells, we employed ImageJ software to analyze the threshold of red color (representing lipid droplets) against a black background. This quantitative approach enables us to accurately measure the area occupied by lipid droplets in the images. We then used ImageJ to count the total number of cells on each coverslip. By dividing the quantified lipid droplet area by the number of cells, we obtained a ratio that provides a quantitative measure of foam cell formation (**page 6, lines 8-13**).

We have added scale bars to all figures to ensure accurate size reference for all images.

2. Individual replicates need to be shown in all figures. Moving away from scale bars will allow for a better assessment of the data quality and rigor

We appreciate the reviewer's suggestion to include individual replicates in our figures. In response, we have revised all relevant figures to display means and standard errors to specifically enhance the assessment of data quality and rigor.

3. What sex were animals used in the in vivo studies? Could the authors please provide separate analysis for the different sexes?

We thank the reviewer for their important question regarding the sex of the animals used in our in vivo studies. We apologize for not clearly stating this information in our original manuscript. To address this, we have clarified that the mice were all male (**page 7, lines 2-11**).

4. Could the authors report the number of independent times experiments were performed, and the number of replicates use in each study?

This has now been clarified in each figure legend.

5. Plaque analysis in the Aortic sinus is unconventional; this site is typically measured as Plaque Area- using a size measurement, instead of a %.

We have now revised our measurements and report them in mm² in our new **Figure 5**.

6. What are the AU that CD68 is measured in? Typically this would be an area measurement or a percentage of CD68+/Plaque area.

We now clarify this in the methods and provide an appropriate reference (**page 9, lines 20-22; page 10, lines 1-3**). AU refers to the consistent fluorescent staining intensity.

7. Figure 6 is not referenced appropriately in the text.

We have corrected this, and now reference the new Figure 7 more appropriately in the text (**page 13, lines 4-14**).

8. Financial disclosures are not clearly stated.

This now clarified further (**page 25, lines 11-13**).

REVIEWERS' COMMENTS:

Reviewer #1 (Remarks to the Author):

The authors have adequately addressed my previous concerns. I believe this paper presents novel information that will be of significant interest to the readers.

Thank you!

Reviewer #2 (Remarks to the Author):

No further comments.

Thank you!

Reviewer #3 (Remarks to the Author):

This is an interesting study. The revised paper addressed any technical concerns that I had regarding the data/rigor of the study. It would be nice to have additional mechanisms, but that is likely outside the scope of this manuscript and I agree with the authors on this point.

Thank you!